# Improving Out-of-Distribution Generalization by Adversarial Training with Structured Priors

**Qixun Wang**[1*]  **Yifei Wang**[2*]  **Hong Zhu**[3]  **Yisen Wang**[1,4†]

[1] Key Lab. of Machine Perception (MoE),
School of Intelligence Science and Technology, Peking University
[2] School of Mathematical Sciences, Peking University
[3] Huawei Noah's Ark Lab
[4] Institute for Artificial Intelligence, Peking University

## Abstract

Deep models often fail to generalize well in test domains when the data distribution differs from that in the training domain. Among numerous approaches to address this Out-of-Distribution (OOD) generalization problem, there has been a growing surge of interest in exploiting Adversarial Training (AT) to improve OOD performance. Recent works have revealed that the robust model obtained by conducting sample-wise AT also retains transferability to biased test domains. In this paper, we empirically show that sample-wise AT has limited improvement on OOD performance. Specifically, we find that AT can only maintain performance at smaller scales of perturbation while Universal AT (UAT) is more robust to larger-scale perturbations. This provides us with clues that adversarial perturbations with universal (low dimensional) structures can enhance the robustness against large data distribution shifts that are common in OOD scenarios. Inspired by this, we propose two AT variants with low-rank structures to train OOD-robust models. Extensive experiments on DomainBed benchmark show that our proposed approaches outperform Empirical Risk Minimization (ERM) and sample-wise AT. Our code is available at `https://github.com/NOVAglow646/NIPS22-MAT-and-LDAT-for-OOD`.

## 1 Introduction

Existing deep learning methods have achieved good performance on visual classification tasks under the same distribution of training sets and test sets. However, when the data distribution of the test set is different from that of the training set, the classification performance of the deep neural networks (DNNs) may decrease sharply [1]. This is mainly because DNNs may capture spurious features such as the background and style information to assist the fast fitting during the training process [2]. However, in real-world scenarios, test data may differ from training data in the background and style information, thus DNNs that rely on unstable spurious features to make predictions will fail. Solving the above problem is known as the out-of-distribution (OOD) generalization.

Another scenario where DNNs may fail is that they are often vulnerable to adversarial examples [3]. Adversarial training (AT) is originally proposed as an effective way to defend against adversarial attacks [4]. Moreover, there is work showing that adversarial training helps to solve the OOD generalization problem because OOD data can be seen as stronger perturbations to some extent [5]. The reason why AT can defend against adversarial attacks meanwhile benefit OOD generalization is that it can make DNNs robust to the interference of spurious features, such as randomly injected noise

---

[*]Equal Contribution.
[†]Corresponding author: Yisen Wang (yisen.wang@pku.edu.cn).

36th Conference on Neural Information Processing Systems (NeurIPS 2022).

(in adversarial examples) or the spurious correlation between labels and background information (in OOD generalization). In other words, AT enables DNNs to make predictions using intrinsic features rather than spurious features.

A potential problem, however, is that existing AT methods ignore the specific design of perturbations when used for solving OOD generalization problems. They usually simply conduct sample-wise AT [6], which only brings limited performance improvement to OOD generalization. The essential reason for the failure of this type of approach is that the perturbations it uses cannot distinguish invariant and spurious features. As a result, it improves the robustness at the expense of the decreasing standard accuracy [7]. Moreover, we empirically find that when adapting Universal AT (UAT [8]) to OOD problems, i.e., conducting AT with domain-wise perturbations, it shows stronger input-robustness when facing larger-scale perturbations compared to the sample-wise AT (see Section 3.2). Since the sample injected with large-scale perturbations can be regarded as OOD samples [5], we draw inspiration from this phenomenon that AT with universal (low-dimensional) structures can be the key to solving OOD generalization. Therefore, we propose to use structured low-rank perturbations related to domain information in AT, which can help the model to filter out background and style information, thus benefiting OOD generalization. We make the following contributions in our work:

- We identify the limitations of sample-wise AT on OOD generalization through a series of experiments. To alleviate this problem, we further propose two simple but effective AT variants with structured priors to improve OOD performances.

- We theoretically prove that our proposed structured AT approach can accelerate the convergence of reliance on spurious features to 0 when using finite-time-stopped gradient descent, thus enhancing the robustness of the model against spurious correlations.

- By conducting experiments on the DomainBed benchmark [9], we demonstrate that our methods outperform ERM and sample-wise AT on various OOD datasets.

## 2 Related Work

**Solving OOD Generalization with AT.** According to [3], the performance of deep models is susceptible to small-scale perturbations injected in the input images, even if these perturbations are imperceptible to humans. Adversarial training (AT) is an effective approach to improve the robustness to input perturbations [4, 10, 11]. However, many recent works have begun to focus on the connection between AT and OOD due to the fact that OOD data can be regarded as one kind of large-scale perturbation. These works seek to exploit the robustness provided by AT to improve OOD generalization. For instance, [6] applied sample-wise AT to OOD generalization. They theoretically found that if a model is robust to input perturbation on training samples, it also generalizes well on OOD data. [5] theoretically established a link between the objective of AT and the OOD robustness. They revealed that the AT procedure can be regarded as a heuristic solution to the worst-case problem around the training domain distribution. Nevertheless, the discussion of [6] and [5] is restricted to the framework of using Wasserstein distance to measure the distribution shift, which is less practical for the real-world OOD setting where domain shifts are diverse. Additionally, they only studied the case of sample-wise AT and did not further investigate the effect of different forms of AT (not sample-wise) on OOD performance. Other works such as [12] focus on the structure design of the perturbations. They used multi-scale perturbations within one sample, but they did not exploit the universal information within one training domain. In our work, we focus on real-world OOD scenarios where there are additional clues lying in the distribution shifts, i.e, the low-rank structures in the spurious features (such as background and style information) across one domain. We further design a low-rank structure in the perturbations to specifically eliminate such low-rank spurious correlations.

**OOD Evaluation Benchmark.** The DomainBed benchmark [9] provides a fair way of evaluating different state-of-the-art OOD methods, which has been widely accepted by the community. By conducting rigorous experiments in a consistent setting, they revealed that many algorithms that claim to outperform previous methods cannot even outperform ERM. Unlike previous works using AT to address OOD generalization, such as [6] and [5], we adopt the Domainbed benchmark for a fair comparison of our approach with existing state-of-the-art methods in this paper.

# 3 Weakness of Sample-wise AT for OOD Generalization

## 3.1 Preliminaries

**Out-of-distribution (OOD) Generalization**. Assuming $x \in \mathcal{X}$ as the random data in the input space $\mathcal{X}$ and $y \in \mathcal{Y}$ as the target random data in the label space $\mathcal{Y}$, we have the predictor $f = w \circ \phi(x)$ where $\phi : \mathcal{X} \to \mathcal{Z}$ denotes the feature extractor and $w : \mathcal{Z} \to \mathcal{Y}$ denotes the classifier.

Now we give the formal definition of the OOD generalization problem. We have a set of $m$ training domains $\mathcal{E} = \{E_1, E_2, ..., E_m\}$, where each domain $E_e$ is characterized by a input dataset $E_e := \{(x_i^e, y_i^e)\}_{i=1}^{n_e}$ containing $n_e$ i.i.d input samples drawn from the distribution of $\mathcal{P}_e$, and a test domain $E_{m+1}$ with data following the distribution of $\mathcal{P}_{te}$, where $\mathcal{P}_{te} \neq \mathcal{P}_i,\ i = 1, 2, ..., m$. $\mathcal{L} : \mathcal{X} \to \mathbb{R}^+$ denotes the loss function. The ultimate goal of OOD generalization is to find an optimal predictor $f$ that minimizes the risk on the unseen test domain:

$$\min_f \mathbb{E}_{(x,y) \sim \mathcal{P}_{te}(x,y)}[\mathcal{L}(f(x), y)]. \tag{1}$$

**Adversarial Training (AT)**[3]. According to [4], AT can be expressed as the following optimization problem:

$$\min_f \mathbb{E}_{(x,y) \sim \mathcal{P}(x,y)}[\max_{\delta \in \mathcal{S}} \mathcal{L}(f(x + \delta), y)] \ \text{ s.t. } \|\delta\|_p \leq \epsilon, \tag{2}$$

where $\delta \in \mathcal{S}$ is the random injected perturbation with $l_p$ norm bounded by $\epsilon$. The inner maximization problem can be optimized by fast gradient sign method (FGSM [13]), a simple one-step scheme:

$$x = x + \epsilon \text{sgn}(\nabla_x \mathcal{L}(f(x), y)), \tag{3}$$

where $\text{sgn}(\cdot)$ is the sign function, or by projected gradient descent (PGD [4]), a more powerful multi-step variant:

$$x^{t+1} = \prod_{\mathcal{S}}(x^t + \gamma \text{sgn}(\nabla_x \mathcal{L}(f(x), y))), \tag{4}$$

where $\prod_{\mathcal{S}}$ is the projection operator onto the set $\mathcal{S}$, $\gamma$ is the step size and $t$ denotes the iteration.

## 3.2 Weakness of AT for OOD Generalization

We now highlight some weaknesses of sample-wise AT for OOD generalization based on a series of empirical evidence. We first conduct a toy experiment on the DomainBed benchmark [9] to evaluate the OOD performance of AT. We run ERM and AT on four OOD datasets: PACS [14], OfficeHome [15], VLCS [16], and NICO [17] with a fixed set of hyperparameters (detailed experimental settings can be found in Appendix C.1). The results are shown in Table 1. We can see that the improvement of OOD performance by AT is limited with an average improvement of only 0.1%.

Table 1: Test accuracy (%) on four OOD datasets on DomainBed benchmark with a fixed set of hyperparameters. The improvement of AT is marginal.

| Algorithm | PACS | OfficeHome | VLCS | NICO | avg |
|---|---|---|---|---|---|
| | | Datasets | | | |
| ERM | $79.7 \pm 0.0$ | $59.6 \pm 0.0$ | $74.4 \pm 1.0$ | $\mathbf{70.7 \pm 1.0}$ | 71.1 |
| AT | $\mathbf{81.5 \pm 0.4}$ | $\mathbf{59.9 \pm 0.4}$ | $\mathbf{75.3 \pm 0.7}$ | $68.2 \pm 2.2$ | $\mathbf{71.2}$ |

We further investigate the reason behind the limitations of performance improvements on OOD datasets of AT. Although previous works have revealed that the robust features obtained by AT can improve OOD generalization ([6] [5] [18]), we find that sample-wise AT only tolerates small-scale perturbations. Thus, we design an experiment on NICO dataset with multiple scales of perturbations. The scale is calculated with the $l_2$ norm of the perturbation matrix (experiment details are shown in Appendix C.1). As shown in Figure 1, AT suffers severe performance degradation when using large perturbations. This provides clues to understanding the failure of AT in OOD scenarios. The

---

[3]For simplicity, we denote 'AT' for sample-wise AT by default in the rest of the paper.

distribution shifts in OOD data usually have much larger scales than the invisible perturbations commonly used in AT. Hence, AT methods designed for small perturbations cannot handle these large-scale domain shifts that often appear in OOD data. However, our experiment shows that this problem can be alleviated by adapting universal AT (UAT [8]) to the OOD setting, i.e., using a perturbation for each domain.

Figure 1 shows that UAT has better generalization performance than AT when the perturbation norm is large. There are two empirical explanations for this: First, the background and style information usually have a low-rank structure, such as the grassland and snowfield that have recurring parts. Second, similar spurious features often appear within one specific domain, such as PACS [14] and VLCS [16] datasets. As stated in [8], the universal perturbation lies in a low dimensional space. Hence using universal (domain-wise) perturbations will help to resist such low-rank shifts and improve the robustness of the model.

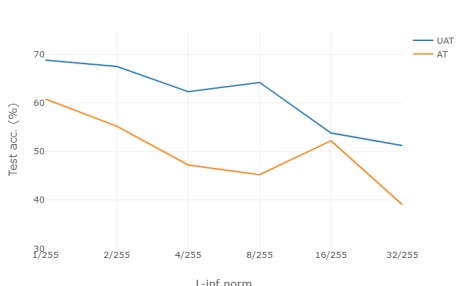

Figure 1: Test accuracy (%) of AT, UAT, and ERM on NICO dataset. $l_\infty$ norm is used here.

Inspired by this, we proposed two new AT variants with more sophisticated low-rank structures on different dimensions to improve OOD generalization in the next section.

## 4  The Proposed Structured AT Method

In order to construct low-rank structured perturbations, we start by analyzing the structure of sample-wise perturbations. Assume that each input data $x$ has a shape of $N \times N \times C$. $N$ is the size of the input image and $C$ is the number of channels. For simplicity, we assume $C = 3$. We reparameterize the sample-wise perturbations as a series of 2-D matrices $\{D_1^1, D_2^1, ..., D_m^1\}$, $\{D_1^2, D_2^2, ..., D_m^2\}$, $\{D_1^3, D_2^3, ..., D_m^3\}$ where $D_e^c \in R^{n_e \times N^2}$ denotes the perturbations in the $e$-th domain for the input channel $c$, $n_e$ is the number of the samples in the domain $E_e$, and $m$ is the number of domains. The $i$-th row of $D_e^c$ represents the $c$-th channel of the $i$-th sample in the domain $E_e$ (see the first column in Figure 2 for illustration). By such reparameterization, it is natural to find that there are two orientations to reduce the rank of the perturbations:

1. **Along the dimension of the number of samples** (along the red arrow in the upper left corner of Figure 2). This corresponds to reducing the number of the perturbations used within one domain.

2. **Along the dimension of the input scale** (along the blue arrow in the upper left corner of Figure 2). This corresponds to reducing the rank of the perturbation used for a specific input sample.

In the following parts, we propose two AT variants with structured priors that reduce the rank in these two directions.

### 4.1  MAT: Adversarial Training with Combinations of Multiple Perturbations

In this part, we propose domain-wise Multiple-perturbation Adversarial Training (MAT). It aims to conduct rank minimization along the dimension of the number of samples. Instead of using sample-wise perturbations, MAT constructs a combination of multiple perturbations and shares this mixed perturbation within a domain. Specifically, we choose to train the linear combination of $k$ perturbations for each domain $E_e$ to conduct AT. Here $k$ is a hyperparameter and $k$ is far less than the number of samples in domain $E_e$. The optimization problem can be reformulated as:

$$\min_f \sum_e \mathbb{E}_{(x,y)\sim\mathcal{P}_e(x,y)}[\mathcal{L}(f(x + \delta^e), y)], \tag{5}$$

$$\text{s.t. } \delta^e = \sum_{i=1}^k \alpha_i^{e*}\delta_i^{e*}, \ \|\delta_i^{e*}\|_p \le \epsilon, \ \sum_{i=1}^k \alpha_i^{e*} = 1, \ \alpha_i^{e*} \ge 0 \text{ for } i = 1, 2, ..., k, \tag{6}$$

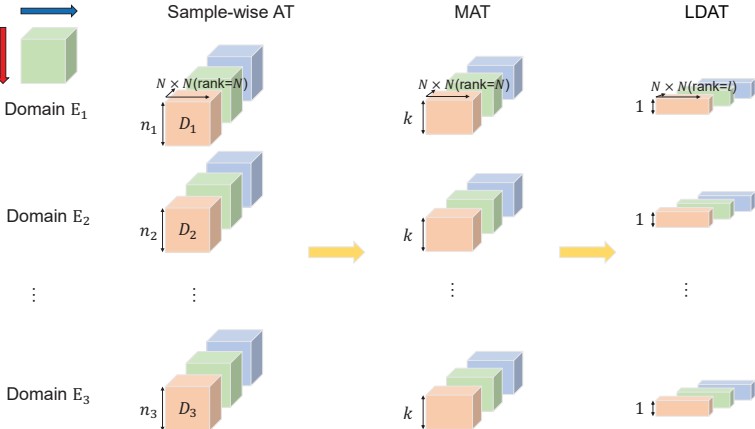

Figure 2: Illustration of how our proposed structured AT reduces the rank of the perturbations comparing to sample-wise AT. The left column shows how we reparameterize sample-wise AT. The perturbations are segmented by domains. A block represents the $n_e$ perturbations injected in a channel of the samples in domain $E_e$. This figure shows the case where the input image has three channels (RGB). The red and the blue arrow in the upper left shows the two orientations to reduce the rank of the perturbations, i.e, along the dimension of the total number of the perturbations and along the dimension of the rank of a single perturbation, respectively. The mid column illustrates that MAT reduces the number of the perturbations used for domain $E_e$ from $n_e$ to $k$ ($k \ll n_e$). The right column shows that LDAT further reduces the number of the perturbations from $k$ to 1. Moreover, it reduces the rank of a specific perturbation from $N$ to $l$ ($l \ll N$).

where

$$\alpha_i^{e*}, \ \delta_i^{e*} = \underset{\alpha_i^e, \ \delta_i^e}{\operatorname{argmax}} \mathbb{E}_{(x,y) \sim \mathcal{P}_e(x,y)} [\mathcal{L}(f(x + \sum_{i=1}^{k} \alpha_i^e \delta_i^e), y)]. \tag{7}$$

Here $e$ denotes the subscript of a training domain and $\alpha_i^e$ is the weight that can be learned for each perturbation $\delta_i^e$. The detailed training procedure of MAT is in Algorithm 1. We first initialize $k$ perturbations $\delta_i^e$ and their correspondent coefficients $\alpha_i^e$ for each training domain $e$ with Gaussian noise. Then we transform $\delta_i^e$ and $\alpha_i^e$ to make sure $\sum_{i=1}^{k} \alpha_i^e = 1$, $\alpha_i^e \geq 0$, and $||\delta_i^e||_2 \leq \epsilon$. For the inner maximization, we conduct a one-step gradient ascent to optimize $\delta_i$ and $\alpha_i$.

MAT works as a low-rank version of sample-wise AT. In sample-wise AT, we maintain $n_e$ perturbations for each domain $E_e$, where $n_e$ is the number of training samples in domain $E_e$. As for MAT, it reduces the number of perturbations available to samples from $n_e$ to $k$ and obtains low-rank structures (see the third column in Figure 2 for illustration). Therefore, it fulfills rank reduction along the sample-number dimension.

## 4.2 LDAT: Adversarial Training with Low-rank Decomposed Perturbations

Based on MAT, we further propose Adversarial Training with Low-rank Decomposed perturbations (LDAT). Analogous to MAT, LDAT still shares one perturbation in a specific domain. Moreover, LDAT imposes a low-rank constraint on the perturbation itself, which corresponds to the dimension of the input scale. Technically, we obtain the domain-wise low-rank perturbation matrix $\delta \in \mathcal{R}^{N \times N \times C}$ by multiplying two matrices: $\delta = AB$. Here $A \in \mathcal{R}^{N \times l \times C}$ and $B \in \mathcal{R}^{l \times N \times C}$ where $l$ is a hyperparameter and $l \ll N$. Since $\operatorname{rank}(AB) \leq \operatorname{rank}(A)$ and $\operatorname{rank}(AB) \leq \operatorname{rank}(B)$ hold for arbitrary matrices $A$, $B$, we have $\operatorname{rank}(\delta) \leq l$. Therefore LDAT reduces the rank of the perturbation from a large value $N$ to a relatively small value $l$ (see the last column in Figure 2 for illustration).

**Algorithm 1** Detailed Training Procedure of MAT

**Input:**
    Labeled training data of $m$ domains $E_1, ..., E_m$, where $E_e := \{(x_i^e, y_i^e)\}_{i=1}^{n_e}$,
    number of the perturbations to be combined $k$, perturbation weight $\alpha$ learning rate $\eta$,
    FGSM step size $\gamma$, perturbation radius $\epsilon$,
    number of training epochs $T$, learning rate for model parameters $r$, batch size $b$.

**Output:**
    Updated model $f_\theta$ with parameter $\theta$.

1: Randomly initiate $\theta$, perturbation $\delta_i^e$, weight $\alpha_i^e$ such that $\sum_{i=1}^k \alpha_i^e = 1$, $\alpha_i^e \geq 0$, $\|\delta_i^e\|_2 \leq \epsilon$,
    $\forall i \in \{1, ..., k\}$ and $\forall e \in \{1, ..., m\}$.
2: **for** iterations in $1, 2, ..., T$ **do**
3:     **for** $e$ in $1, 2, ..., m$ **do**
4:         Randomly select batch $\mathcal{B}^e = \{(x_u^e, y_u^e)\}_{u=1}^b$ from domain $E_e$.
5:         Compute the adversarial sample: $x_u^{e'} = x_u^e + \sum_{j=1}^k \alpha_j^e \delta_j^e$, $\forall u \in \{1, ..., b\}$
6:         Update $\delta^e$ by $\delta_i^e \leftarrow \delta_i^e + \gamma \frac{1}{b} \sum_{u=1}^b \nabla_{\delta_i^e} \mathcal{L}(f_\theta(x_u^{e'}), y_u^e)$, $\forall i \in \{1, ..., k\}$, $\forall u \in \{1, ..., b\}$.
7:         Update $\alpha_i^e$ by $\alpha_i^e \leftarrow \alpha_i^e + \eta \frac{1}{b} \sum_{u=1}^b \nabla_{\alpha_i^e} \mathcal{L}(f_\theta(x_u^{e'}), y_u^e)$, $\forall i \in \{1, ..., k\}$, $\forall u \in \{1, ..., b\}$.
8:         Project $\delta_i^e$ to the $l_2$ ball of radius $\epsilon$.
9:         Compute the adversarial sample: $x_u^{e'} = x_u^e + \sum_{j=1}^k \alpha_j^e \delta_j^e$, $\forall u \in \{1, ..., b\}$
10:       Update model parameter: $\theta \leftarrow \theta - r \frac{1}{b} \sum_{u=1}^b \nabla_\theta \mathcal{L}(f_\theta(x_u^{e'}), y_u^e)$, $\forall u \in \{1, ..., b\}$.
11:     **end for**
12: **end for**

---

The formal definition of the LDAT objective is:

$$\min_f \sum_e \mathbb{E}_{(x,y) \sim \mathcal{P}_e(x,y)}[\mathcal{L}(f(x + \delta^e), y)], \text{ s.t. } \delta^e = A^{e*} B^{e*}, \|\delta^e\|_p \leq \epsilon, \tag{8}$$

where

$$A^{e*}, B^{e*} = \underset{A^e, B^e}{\operatorname{argmax}} \mathbb{E}_{(x,y) \sim \mathcal{P}_e(x,y)}[\mathcal{L}(f(x + A^e B^e), y)], \ A^e \in \mathcal{R}^{N \times l \times C}, \ B^e \in \mathcal{R}^{l \times N \times C}. \tag{9}$$

We provide the detailed training procedure of LDAT in Appendix D due to the space limitation of the main text. In comparison to MAT, LDAT reduces the number of perturbations available to the samples in a domain from $k$ to 1. In addition, it reduces the rank of the perturbation for a single channel of a sample from $N$ to $l$.

### 4.3 Theoretical Analysis

In this part, we theoretically explain why the domain-wise perturbation proposed in MAT and LDAT can help to improve the robustness of the model against spurious correlations following [2] and [19]. In general, we prove that MAT and LDAT can prevent the model from relying more on spurious features to make predictions as the spurious correlations in the training data increase. Consequently, the model trained with MAT or LDAT will generalize better on OOD data.

**Notations.** Let $x \in \mathcal{X}$ denote the random data in the input space $\mathcal{X}$ and let $y \in \mathcal{Y}$ denote the target random data in label space $\mathcal{Y}$. For simplicity, let $\mathcal{Y} \in \{1, -1\}$ in this section. Let $\mathbb{D}$ denote an underlying class of distributions over $\mathcal{X} \times \mathcal{Y}$. Let $x_{inv}$ and $x_{sp}$ denote the invariant features and the spurious features respectively. Also for simplicity, assume that there exists an identity mapping $\Phi : \mathcal{X}_{inv} \times \mathcal{X}_{sp} \to \mathcal{X}$ such that each $\mathcal{D} \in \mathbb{D}$ is induced by a distribution over $\mathcal{X}_{inv} \times \mathcal{X}_{sp}$ (so $x$ can be denoted as $x = (x_{inv}, x_{sp})$). Let $x_{sp}$ take values in $\{+\beta, -\beta\}$ for some $\beta > 0$.

**A Simple OOD Task.** Consider a simple OOD task where we have two training domains representing the grass and desert backgrounds respectively. Both domains have two classes: the cow class and the camel class. In the grass/desert domain, the cow/camel class predominates. During test time, the correlation between the labels and the background flips. We can abstract this cow-camel dataset into the following model: a training dataset $\mathcal{S}$ with four groups of data points drawn from the four quadrants of the feature space $\{-1, +1\} \times \{-\beta, +\beta\}$ respectively (shown in Figure 3). We set the invariant features $x_{inv} = y$ and the spurious features $x_{sp}$ to be $y\beta$ with probability $p \in [0.5, 1)$ and

$-y\beta$ with probability $1 - p$. Note that $p$ measures the intensity of spurious correlations in a certain environment. When $p = 0.5$, there are no correlations between the labels and the spurious features.

Consider a linear classifier $h(x) = w_{inv}x_{inv} + w_{sp}x_{sp}$. Following [19], let us consider MAT/LDAT trained with gradient descent algorithm stopped in finite time $t$. In order to characterize the dependence of the model on spurious features during the training process, we investigate the convergence rate of $\frac{w_{sp}(t)\beta}{|w_{inv}(t)x_{inv}|}$ to 0 on the above dataset, which denotes the ratio between the output of the spurious component to that of the invariant component. We prove that after adding the domain-wise perturbations in finite-time-stopped gradient descent, the lower bound of the convergence rate of this ratio does not increase monotonically with $p$. Hence, the model will not learn a large prediction weight based on spurious features even if the spurious correlation is strong ($p$ is large).

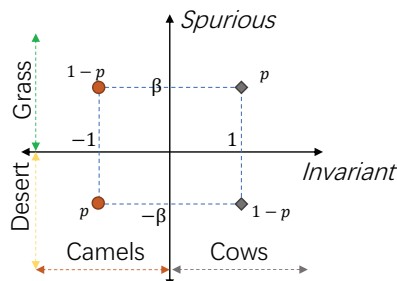

Figure 3: Illustration of the simple OOD task.

In the following theorem, we denote the domain-wise perturbation in MAT/LDAT as $\delta$. Theorem 4.1 applies to both MAT and LDAT since they both use domain-wise perturbations. See Appendix A for a formal statement and full proof of Theorem 4.1.

**Theorem 4.1.** *(informal) Let $\mathcal{H}$ be the set of linear classifiers $h(x) = w_{inv}(t)x_{inv} + w_{sp}(t)x_{sp}$. Consider the above 2-D OOD dataset $\mathcal{S}$. Assume that the empirical distribution of $x_{inv}$ given $x_{sp} \cdot y > 0$ is identical to the empirical distribution of $x_{inv}$ given $x_{sp} \cdot y < 0$. $\delta$ is the optimal perturbation obtained by optimizing the object in Eq. (7) or Eq. (9). Let $w_{inv}(t)x_{inv} + w_{sp}(t)x_{sp}$ be initialized to the origin, and trained with MAT/LDAT to minimize the exponential loss on $\mathcal{S}$. Then, for any $(x, y) \in \mathcal{S}$, we have:*

$$\Omega(\mathbb{E}_{(x_{inv},y)\sim\mathcal{D}_{inv}}[\frac{\frac{1}{\beta+\delta y}\ln[\frac{c_1+p}{c_2+p^{\frac{1}{2}-\epsilon}(1-p)^{\frac{1}{2}+\epsilon}}]}{M\ln(t+1)}]) \leq \frac{w_{sp}(t)\beta}{|w_{inv}(t)x_{inv}|}, \tag{10}$$

*where $\epsilon := \frac{\delta y}{2\beta}$ is a real number close to 0, $c_1 := \frac{2(2M(1+\delta)-1)}{(\beta+\delta y)^2}$, $c_2 := \frac{2(2M(1+\delta)-1)}{(\delta y+\beta)^{\frac{3}{2}-\epsilon}(\beta-\delta y)^{\frac{1}{2}+\epsilon}}$. $M = \max_{x\in\mathcal{S}} \hat{w} \cdot x$ denotes the maximum value of the margin of the max-margin classifier $\hat{w}$ on $\mathcal{S}$. $\Omega(\cdot)$ is the lower bound of a given function within a constant factor. Therefore, the lower bound of the convergence rate does not increase monotonically with $p$ under the condition that $2\epsilon c_1 + c_2 + \frac{3}{4} + \frac{3}{2}\epsilon < 0$.*

To sum up, since we can prevent this lower bound from growing monotonically with $p$, we accelerate the convergence rate of $\frac{w_{sp}(t)\beta}{|w_{inv}(t)x_{inv}|}$ to 0 when there is stronger spurious correlation (larger $p$). Recall that the ratio $\frac{w_{sp}(t)\beta}{|w_{inv}(t)x_{inv}|}$ reflects the degree of reliance on spurious features. Therefore, faster convergence of this ratio to 0 (smaller lower bound) means that the model will end up relying less on the spurious correlations within a finite training time. In other words, the OOD robustness can be enhanced by using domain-wise perturbations.

**Remark.** Here, we demonstrate that MAT and LDAT show stronger OOD robustness compared to ERM. We compare the result in Theorem 4.1 to that in Theorem 2 of [2]. The full statement of the Theorem 2 in [2] is in Appendix B. According to the Theorem 2 in [2], even if the max-margin classifier does not rely on $x_{sp}$ for any level of spurious correlation $p \in [0.5, 1)$, ERM trained by gradient descent stopped in finite time still fails to avoid using spurious features. Moreover, when conducting ERM with finite-time-stopped gradient descent, the lower bound of the convergence rate of $\frac{w_{sp}(t)\beta}{|w_{inv}(t)x_{inv}|}$ to 0 is

$$\Omega(\frac{\ln\frac{c+p}{c+\sqrt{p(1-p)}}}{M\ln t}) \leq \frac{w_{sp}(t)\beta}{|w_{inv}(t)x_{inv}|}, \tag{11}$$

where $c := \frac{2(2M-1)}{\beta^2}$, and $M$ follows the definition in Theorem 4.1. This lower bound grows monotonically with $p$, thus ERM will have slower convergence for larger spurious correlations.

However, with domain-wise perturbations, we can modify the lower bound so that it does not increase monotonically with the spurious correlation $p$. Thus, we can draw the conclusion that using a perturbation for each domain is helpful to reduce dependence on spurious features compared to ERM.

## 5 Experiments

### 5.1 Experimental Setup

We conduct experiments on the DomainBed benchmark [9], a testbed for OOD generalization that implements consistent experimental protocols across various approaches to ensure fair comparisons. We evaluate on PACS [14], OfficeHome [15], VLCS [16], NICO [17], and Colored MNIST [1]. There are several changes in our experimentation setting comparing to DomainBed:

1. **Backbone Network.** We use ResNet-18 [20] as our backbone for datasets excluding Colored MNIST instead of ResNet-50 used in [9] for efficiency.

2. **Hyperparameter Search Space.** We use a smaller hyperparameter search space than [9]. We conduct a random search of 8 trials for PACS, OfficeHome, and VLCS while 6 trials for NICO and Colored MNIST in the hyperparameter search space, instead of 20 trials adopted in [9] for feasibility. See Appendix C.2 for more details.

**Model Selection Strategy.** Since hyperparameter choice has a significant impact on the OOD performance, it is critical to use appropriate model selection method. For PACS, OfficeHome, and VLCS datasets, we use training-domain validation proposed in [9] since it is more in line with the OOD scenario. For NICO, we adopt OOD validation following [21]. For Colored MNIST, we use test-domain validation [9] since it can enlarge the gaps in OOD performance among the algorithms while the gap induced by training-domain validation on Colored MNIST is marginal.

**Hyperparameters for MAT and LDAT.** To retain low-rank structures in perturbations, we set the upper bound of the search space of the perturbation number $k$ in MAT to be 20. Similarly, the upper bound of the rank of the perturbation used in LDAT $l$ is 20. Specifically, the search space of $k$ and $l$ is $\{5, 10, 15, 20\}$ (except on CMNIST, where $k \in [5, 20]$ and $l \in [10, 20]$). The complete setup of the hyperparameters for MAT and LDAT is provided in Appendix C.2.

### 5.2 OOD Performance on Benchmark datasets.

Table 2 summarizes the results on the five OOD datasets. The results of other approaches for PACS, OfficeHome, NICO, and Colored MNIST datasets are adopted from [21]. The results on VLCS of other algorithms are missing (denoted as "-") because [21] does not experiment on this dataset.

**Comparison with ERM and Sample-wise AT.** From Table 2, we observe that both MAT and LDAT outperform ERM (on both our runs and the results in [21]) and AT on average. In particular, MAT achieves consistently better results than ERM on all five datasets. Additionally, the average performance of AT is worse than ERM, which is consistent with our observations in Section 3.2.

**Comparison with Existing State-of-the-Art Approaches.** Although the results from [21] use a different training protocol from ours: they use a larger search space and 20 random search for the hyperparameter combinations, the comparison between ERM ([21]) and ERM (our runs) indicates that their corresponding performances are close. A similar comparison has been made in [22]. We find that MAT outperforms all previous algorithms and LDAT ranked fourth among all methods, merely after VREx [23] and IRM [1] (see avg[1] in Table 2). And even when excluding Colored MNIST (toy example), our methods still outperform ERM by $0.4 \sim 1.4\%$, whereas other methods show no improvement over ERM (see avg[3]). From these results, we can see that the promotion of our proposed methods is higher than the other works, and our methods clearly outperform ERM. We also extend our evaluation to compare with adversarial augmentation based method [5] in Appendix C.3. Single-training domain generalization experiments are shown in Appendix C.4, which shows that our methods can maintain OOD performance without the reliance on multi-source training data.

**Comparison between MAT and LDAT.** From Table 2 we can see that MAT outperforms LDAT on average. Since LDAT reduces the number of the perturbations used in a domain from $k$ to 1 (shown in Figure 2), LDAT can be regarded as a low-rank version of MAT. This indicates that the oversimplified perturbations may be less effective than the ones maintaining some flexibility.

Table 2: Test accuracy (%) on OOD datasets within DomainBed benchmark using ResNet-18. Here "avg[1]" denotes the average accuracy on PACS, OfficeHome, NICO, CMNIST datasets and "avg[2]" denotes the average accuracy on all five datasets. "avg[3]" denotes the average accuracy on the other three datasets except for CMNIST and VLCS. "avg[4]" denotes the average accuracy on the other four datasets except for CMNIST. The best results are in **bold**.

| Algorithm | Datasets | | | | | avg[1] | avg[2] | avg[3] | avg[4] |
| | PACS | OfficeHome | VLCS | NICO | CMNIST | | | | |
|---|---|---|---|---|---|---|---|---|---|
| ERM (Our runs) | $81.7 \pm 0.3$ | $62.1 \pm 0.1$ | $74.4 \pm 1.0$ | $73.2 \pm 1.9$ | $28.1 \pm 1.5$ | 61.3 | 63.9 | 72.3 | 72.9 |
| AT (Our runs) | $81.5 \pm 0.3$ | $62.1 \pm 0.3$ | $\mathbf{76.2 \pm 0.3}$ | $69.7 \pm 1.6$ | $29.1 \pm 1.5$ | 60.6 | 63.7 | 71.1 | 72.4 |
| ERM[21] | $81.5 \pm 0.0$ | $63.3 \pm 0.2$ | - | $71.4 \pm 1.3$ | $29.9 \pm 0.1$ | 61.5 | - | 72.1 | - |
| RSC[24] | $\mathbf{82.8} \pm 0.4$ | $62.9 \pm 0.4$ | - | $69.7 \pm 0.3$ | $28.6 \pm 1.5$ | 61.0 | - | 71.8 | - |
| MMD[25] | $81.7 \pm 0.2$ | $63.8 \pm 0.1$ | - | $68.3 \pm 1.8$ | $50.7 \pm 0.1$ | 66.1 | - | 71.3 | - |
| SagNet[26] | $81.6 \pm 0.4$ | $62.7 \pm 0.4$ | - | $69.3 \pm 1.0$ | $30.5 \pm 0.7$ | 61.0 | - | 71.2 | - |
| CORAL[27] | $81.6 \pm 0.6$ | $63.8 \pm 0.3$ | - | $68.3 \pm 1.4$ | $30.0 \pm 0.5$ | 61.0 | - | 71.2 | - |
| IRM[1] | $81.1 \pm 0.3$ | $63.0 \pm 0.2$ | - | $67.6 \pm 1.4$ | $60.2 \pm 2.4$ | 68.0 | - | 70.6 | - |
| VREx[23] | $81.8 \pm 0.1$ | $63.5 \pm 0.1$ | - | $71.0 \pm 1.3$ | $56.3 \pm 1.9$ | 68.2 | - | 72.1 | - |
| GroupDRO[28] | $80.4 \pm 0.3$ | $63.2 \pm 0.2$ | - | $71.8 \pm 0.8$ | $32.5 \pm 0.2$ | 62.0 | - | 71.8 | - |
| DANN[29] | $81.1 \pm 0.4$ | $62.9 \pm 0.6$ | - | $68.6 \pm 1.1$ | $24.5 \pm 0.8$ | 59.3 | - | 70.9 | - |
| MTL[30] | $81.2 \pm 0.4$ | $62.9 \pm 0.2$ | - | $70.2 \pm 0.6$ | $29.3 \pm 0.1$ | 60.9 | - | 71.4 | - |
| Mixup[31] | $79.8 \pm 0.6$ | $63.3 \pm 0.5$ | - | $66.6 \pm 0.9$ | $27.6 \pm 1.8$ | 59.3 | - | 69.9 | - |
| ANDMask[32] | $79.5 \pm 0.0$ | $62.0 \pm 0.3$ | - | $72.2 \pm 1.2$ | $27.2 \pm 1.4$ | 60.2 | - | 71.2 | - |
| MLDG[33] | $73.0 \pm 0.4$ | $52.4 \pm 0.2$ | - | $51.6 \pm 6.1$ | $32.7 \pm 1.1$ | 52.4 | - | 59.0 | - |
| MAT (Our work) | $82.3 \pm 0.5$ | $\mathbf{64.5 \pm 2.1}$ | $74.6 \pm 0.8$ | $74.2 \pm 1.5$ | $\mathbf{65.4 \pm 8.1}$ | **71.6** | **72.2** | **73.7** | **73.9** |
| LDAT (Our work) | $82.6 \pm 0.5$ | $61.0 \pm 0.9$ | $75.3 \pm 0.3$ | $\mathbf{74.4 \pm 1.6}$ | $52.5 \pm 5.4$ | 67.6 | 69.1 | 72.7 | 73.3 |

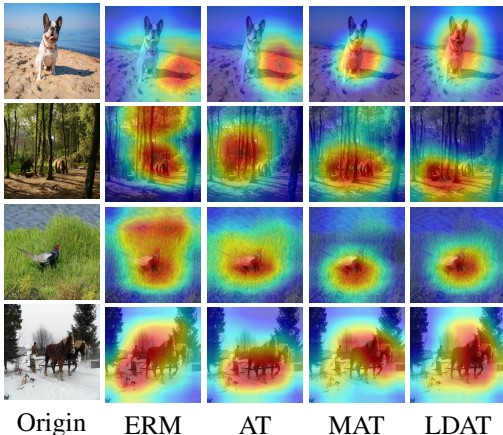

Origin ERM AT MAT LDAT

Figure 4: The pixel attention heatmap of ERM, AT, MAT and LDAT on NICO dataset. The redder part indicates that the model relies more on this part to make predictions.

Since both MAT and LDAT outperform most existing state-of-the-art methods and they both exploit low-rank structures, these two methods mutually corroborate the effectiveness of low-rank structure for OOD generalization. The respective advantages of the two methods are as follows:

- MAT is a complex and high-rank version of LDAT, which has a stronger ability to describe more complex spurious background information. As shown by the attention heatmap in Figure 4, MAT can better capture the object than LDAT when faced with a more complex background (the example in the second row of Figure 4).

- LDAT costs less memory than MAT during the training process, although there is no significant difference in training time between the two methods. When the memory is limited, LDAT is preferred.

### 5.3 Empirical Understanding

**Visualization.** To empirically show that MAT and LDAT can reduce the reliance on spurious features, we visualize the pixel attention heatmap of ERM, AT, MAT, and LDAT on NICO dataset using GradCam [34]. It reflects the contribution of different components of the feature map to the prediction results. We pick the model with the best performance for each method. The results in Figure 4 indicate that the model trained by MAT and LDAT focuses more on the object itself, while ERM and AT adopt the background information that spuriously correlates to the class to make predictions.

**Parameter Analysis.** The number of the perturbations $k$ used in a domain in MAT and the rank of the perturbation $l$ in LDAT are two key hyperparameters. We conduct further experiments to analyze the impact on the performances of $k$ and $l$. We adopt a fixed set of parameters except for $k$ and $l$ and evaluate on PACS dataset. The results in Figure 5 show that MAT and LDAT are able to keep their performances over ERM as long as $k$ and $l$ are far less than the number of the samples $N$. Additionally, we can observe from the trend that when $k$ and $l$ are too small ($= 5$), the performances degenerate. This implies the oversimplified structures of the perturbations can be less effective for generalization. When $k$ and $l$ take larger value (about 1000), the performance will drop (see Table 3). For the selection of optimal parameters (range), we observe that the parameters that are good on one dataset also work well on others (see Table 6 in the appendix), so in practice we adopt the strategy of searching for the optimal parameters roughly on one dataset and then applying them to other datasets. Additional analysis on the impact of the learning rate for the perturbations is in Appendix C.2.

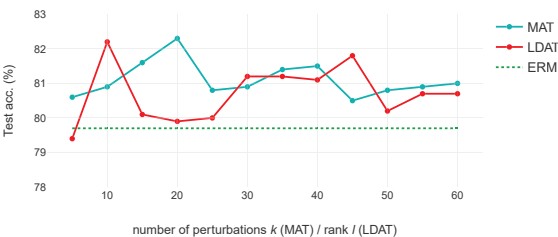

Figure 5: The performance at different values of the number of perturbations to combined within one domain $k$ (in MAT) and the rank of the perturbation in a domain $l$ (in LDAT).

Table 3: The test accuracy (%) on CMNIST with different $k$ and $l$.

| Algorithm | $k \in [5, 20], l \in [10, 20]$ | $k$ or $l = 200$ | $k$ or $l = 500$ | $k$ or $l = 1000$ |
|---|---|---|---|---|
| MAT | $65.4 \pm 8.1$ | $34.9 \pm 20.2$ | $25.6 \pm 8.5$ | $23.4 \pm 10.8$ |
| LDAT | $52.5 \pm 5.4$ | $24.9 \pm 8.9$ | $19.0 \pm 6.6$ | $10.3 \pm 0.1$ |

## 6 Conclusion

In this work, we empirically revealed the limitations of sample-wise AT on OOD tasks. Due to the lack of constraints on the perturbation and the utilization of domain features, sample-wise AT fails to generalize well when facing large-scale perturbations which is close to the real-world OOD scenarios. We further proposed two AT variants with structured priors, named MAT and LDAT, which add low-rank perturbations to improve model's robustness against the distribution shift of spurious correlations. We theoretically proved the domain-wise perturbations used in MAT and LDAT can benefit OOD generalization, and validated the effectiveness of the proposed methods on OOD tasks through a series of experiments on Domainbed benchmark.

## Acknowledgment

Qixun Wang is partially supported by the State Key Development Program Grand (No. 2020YFB1708002). Yisen Wang is partially supported by the NSF China (No. 62006153), Project 2020BD006 supported by PKU-Baidu Fund, Open Research Projects of Zhejiang Lab (No. 2022RC0AB05), and Huawei Technologies Inc.

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
