# A  Proof of Theorem 4.1

**Theorem** 4.1 **(formal)** Let $\mathcal{H}$ be the set of linear classifiers $h(x) = w_{inv}(t)x_{inv} + w_{sp}(t)x_{sp}$. Consider any task that satisfies all the constraints in Section 3.1. in [2]. Consider a dataset $\mathcal{S}$ drawn from $\mathcal{D}$ such that the empirical distribution of $x_{inv}$ given $x_{sp} \cdot y > 0$ (denoted as $(x_{inv}, y) \sim \mathcal{D}_{inv}$) is identical to the empirical distribution of $x_{inv}$ given $x_{sp} \cdot y < 0$. $\delta$ is the optimal perturbation obtained by optimizing object (7) or (9). $\delta$ can be seen as a random variable.

Let $w_{inv}(t)x_{inv} + w_{sp}(t)x_{sp}$ be initialized to the origin, and trained with MAT/LDAT with an infinitesimal learning rate to minimize the exponential loss on $\mathcal{S}$. Then, for any $(x, y) \in \mathcal{S}$, we have:

$$\Omega(\mathbb{E}_{(x_{inv}, y) \sim \mathcal{D}_{inv}}[\frac{\frac{1}{\beta + \delta y}\ln[\frac{c_1 + p}{c_2 + p^{\frac{1}{2} - \epsilon}(1 - p)^{\frac{1}{2} + \epsilon}}]}{M \ln(t + 1)}]) \leq \frac{w_{sp}(t)\beta}{|w_{inv}(t)x_{inv}|}, \qquad (12)$$

where $\epsilon := \frac{\delta y}{2\beta}$ is a real number close to 0, $c_1 := \frac{2(2M(1 + \delta) - 1)}{(\beta + \delta y)^2}$, $c_2 := \frac{2(2M(1 + \delta) - 1)}{(\delta y + \beta)^{\frac{3}{2} - \epsilon}(\beta - \delta y)^{\frac{1}{2} + \epsilon}}$. $M = \max\limits_{x \in S} \hat{w} \cdot x$ denotes the maximum value of the margin of the max-margin classifier $\hat{w}$ on $\mathcal{S}$. $\Omega(\cdot)$ is the lower bound of a given function within a constant factor. Therefore, the lower bound of the convergence rate does not increase monotonically with $p$ under the condition that $2\epsilon c_1 + c_2 + \frac{3}{4} + \frac{3}{2}\epsilon < 0$.

*Proof.* For brevity, we use $w_c$ and $w_s$ to represent $w_{inv}$ and $w_{sp}$ respectively. Also, we use $x_c$ to represent $x_{inv}$. We use $x_e$ to represent $x_{sp}$ since the spurious feature is correlated to the environment $E_e$. Let $\mathcal{S}_{min}$ and $\mathcal{S}_{maj}$ denote the subset of datapoints in $\mathcal{S}$ where $x_e \cdot y < 0$ and $x_e \cdot y > 0$ respectively. Let $\mathcal{D}_{inv}$ denote the distribution over $(x_c, y)$ induced by drawing $(x, y)$ uniformly from $\mathcal{S}_{min}$. The corresponding marginal distribution of $y$ is denoted as $\mathcal{D}_y$. By the assumption of the theorem, this distribution would be the same if $x$ was drawn uniformly from $\mathcal{S}_{maj}$. Then, the loss function that is being minimized in this setting corresponds to:

$$\begin{aligned}
\mathcal{L}[f(x), y] &= \mathbb{E}_{(x_c, y) \sim \mathcal{D}_{inv}}[e^{-f(x)y}] \\
&= \mathbb{E}_{(x_c, y) \sim \mathcal{D}_{inv}}[e^{-[w_c x_c + w_s x_e + (w_c + w_s)\delta]y}] \\
&= \mathbb{E}_{(x_c, y) \sim \mathcal{D}_{inv}}[pe^{-[w_c x_c + w_s \beta y + (w_c + w_s)\delta]y} + (1 - p)e^{-[w_c x_c - w_s \beta y + (w_c + w_s)\delta]y}] \\
&= \mathbb{E}_{(x_c, y) \sim \mathcal{D}_{inv}}[e^{-(x_c + \delta)w_c y}[pe^{-(\delta y + \beta)w_s} + (1 - p)e^{(\beta - \delta y)w_s}]]
\end{aligned}$$

The update on $w_s$ can be written as:

$$\begin{aligned}
\Delta w_s &= -\frac{\partial \mathcal{L}[f(x), y]}{\partial w_s} \\
&= -\mathbb{E}_{(x_c, y) \sim \mathcal{D}_{inv}} e^{-(x_c + \delta)w_c y}[-pe^{-(\delta y + \beta)w_s}(\delta y + \beta) + (1 - p)e^{(\beta - \delta y)w_s}(\beta - \delta y)]
\end{aligned}$$

**Proof of bounds on $w_c(t)x_c$.** Using the result of [19] and [2], we get

$$|w_c(t)x_c| \in [0.5\ln(1 + t), 2M\ln(1 + t)]$$

for a sufficiently large $t$ and for all $x \in \mathcal{S}$.

**Proof of the upper bound on $w_s$.** To calculate the lower bound of $w_s$, we prove the upper bound as auxiliary first. Note that $\Delta w_s$ decreases monotonically with $w_s$. Assume that $\beta > |\delta|$ (this is reasonable since the perturbation radius is usually smaller than the scale of the spurious feature). Let $\Delta w_s = 0$, we get

$$w_s = \frac{1}{2\beta}\ln[\frac{p}{1 - p}(\frac{\beta + \delta y}{\beta - \delta y})] =: w_0.$$

Since $\Delta w_s$ decrease monotonically with $w_s$, (which can be inferred from $\frac{\partial \Delta w_s}{\partial w_s} = -e^{-(x_c + \delta)w_c y}[-pe^{-(\delta y + \beta)w_s}(\delta y + \beta)^2 + (1 - p)e^{(\beta - \delta y)w_s}(\beta - \delta y)^2] \leq 0$ ), when $w_s < w_0$, $\Delta w_s > 0$ and when $w_s > w_0$, $\Delta w_s < 0$. As a result, for any system that is initialized at 0, $w_s$ can never cross the point $w_0$. Thus, we get the upper bound of $w_s(t)$:

$$w_s(t) < w_0 = \frac{1}{2\beta}\ln[\frac{p}{1 - p}(\frac{\beta + \delta y}{\beta - \delta y})].$$

**Proof of the lower bound on $w_s$.** We lower bound $w_s$ via the upper bound on $w_s$ as:

$$\Delta w_s > \mathbb{E}_{(x_c,y)\sim\mathcal{D}_{inv}} e^{-(x_c+\delta)w_c y}[p(\delta y+\beta)e^{-(\delta y+\beta)w_s} - (1-p)(\beta-\delta y)[\frac{p(\beta+\delta y)}{(1-p)(\beta-\delta y)}]^{\frac{\beta-\delta y}{2\beta}}]$$

$$= \mathbb{E}_{(x_c,y)\sim\mathcal{D}_{inv}} e^{-(x_c+\delta)w_c y}[p(\delta y+\beta)e^{-(\delta y+\beta)w_s} - [p(\beta+\delta y)]^{\frac{1}{2}-\frac{\delta y}{2\beta}}[(1-p)(\beta-\delta y)]^{\frac{1}{2}+\frac{\delta y}{2\beta}}].$$

Next, using the upper bound on $|w_c(t)x_c|$, we get:

$$\Delta w_s > \mathbb{E}_{(x_c,y)\sim\mathcal{D}_{inv}} e^{-2M\ln(1+t)}e^{-2M\delta\ln(1+t)}[p(\delta y+\beta)e^{-(\delta y+\beta)w_s}$$

$$- [p(\beta+\delta y)]^{\frac{1}{2}-\frac{\delta y}{2\beta}}[(1-p)(\beta-\delta y)]^{\frac{1}{2}+\frac{\delta y}{2\beta}}]$$

$$= \mathbb{E}_{(x_c,y)\sim\mathcal{D}_{inv}} \frac{1}{(t+1)^{2M(1+\delta)}}[p(\delta y+\beta)e^{-(\delta y+\beta)w_s}$$

$$- [p(\beta+\delta y)]^{\frac{1}{2}-\frac{\delta y}{2\beta}}[(1-p)(\beta-\delta y)]^{\frac{1}{2}+\frac{\delta y}{2\beta}}].$$

For brevity, we denote the term $p(\delta y+\beta)e^{-(\delta y+\beta)w_s}$ as $T$ and denote the term $[p(\beta+\delta y)]^{\frac{1}{2}-\frac{\delta y}{2\beta}}[(1-p)(\beta-\delta y)]^{\frac{1}{2}+\frac{\delta y}{2\beta}}$ as $L$. And in the following proof, we omit the expectation marker $\mathbb{E}$ also for simplicity. It is clear that both $T$ and $L > 0$. Here, $T - L > 0$ since $w_s < w_0$. Note that $\Delta w_s = -\frac{\partial \mathcal{L}[f(x),y]}{\partial w_s} = \frac{\partial w_s}{\partial t}$, rearranging this and integrating, we get:

$$\int_0^{w_s} \frac{1}{p(\beta+\delta y)e^{-(\beta+\delta y)w_s}-L}dw_s > \int_0^t \frac{1}{(t+1)^{2M(1+\delta)}}dt,$$

$$\frac{\ln[p(\beta+\delta y)-L]-\ln[p(\beta+\delta y)-e^{(\beta+\delta y)w_s}L]}{(\beta+\delta y)L} > \frac{1}{2M(1+\delta)-1}[1-\frac{1}{(1+t)^{2M(1+\delta)-1}}].$$

Since for a sufficiently large $t$, $1 - \frac{1}{(1+t)^{2M(1+\delta)-1}} > \frac{1}{2}$, we have:

$$\ln[\frac{p(\beta+\delta y)-L}{p(\beta+\delta y)-e^{(\beta+\delta y)w_s}L}] > \frac{(\beta+\delta y)L}{2(2M(1+\delta)-1)},$$

we can further lower bound the right hand side by applying the inequality $x \geq \ln(x+1)$ for positive $x$:

$$\ln[\frac{p(\beta+\delta y)-L}{p(\beta+\delta y)-e^{(\beta+\delta y)w_s}L}] > \ln[\frac{(\beta+\delta y)L}{2(2M(1+\delta)-1)}+1].$$

Thus,

$$\frac{p(\beta+\delta y)-L}{p(\beta+\delta y)-e^{(\beta+\delta y)w_s}L} > 1 + \frac{(\beta+\delta y)L}{2(2M(1+\delta)-1)}.$$

Note that the denominator on the left side of the inequality is greater than 0 since $T - L > 0$. Rearrange this inequality:

$$e^{w_s(\beta+\delta y)} > \frac{1+\frac{(\beta+\delta y)^2 p}{2(2M(1+\delta)-1)}}{1+\frac{(\beta+\delta y)L}{2(2M(1+\delta)-1)}}.$$

Putting $L = [p(\beta+\delta y)]^{\frac{1}{2}-\frac{\delta y}{2\beta}}[(1-p)(\beta-\delta y)]^{\frac{1}{2}+\frac{\delta y}{2\beta}}$ back into the inequality,

$$e^{w_s(\beta+\delta y)} > \frac{1+\frac{(\beta+\delta y)^2 p}{2(2M(1+\delta)-1)}}{1+\frac{(\beta+\delta y)[p(\beta+\delta y)]^{\frac{1}{2}-\frac{\delta y}{2\beta}}[(1-p)(\beta-\delta y)]^{\frac{1}{2}+\frac{\delta y}{2\beta}}}{2(2M(1+\delta)-1)}}$$

$$= \frac{p+\frac{2(2M(1+\delta)-1)}{(\beta+\delta y)^2}}{p^{\frac{1}{2}-\frac{\delta y}{2\beta}}(1-p)^{\frac{1}{2}+\frac{\delta y}{2\beta}}+\frac{2(2M(1+\delta)-1)}{(\delta y+\beta)^{\frac{3}{2}-\frac{\delta y}{2\beta}}(\beta-\delta y)^{\frac{1}{2}+\frac{\delta y}{2\beta}}}}.$$

Let $c_1 := \frac{2(2M(1+\delta)-1)}{(\beta+\delta y)^2}$, $c_2 := \frac{2(2M(1+\delta)-1)}{(\delta y+\beta)^{\frac{3}{2}-\epsilon}(\beta-\delta y)^{\frac{1}{2}+\epsilon}}$, $\epsilon := \frac{\delta y}{2\beta}$.

Put the expectation mark back into this inequality, finally, we get the lower bound on $w_s$:

$$w_s \geq \mathbb{E}_{(x_c,y)\sim\mathcal{D}_{inv}}[\frac{1}{\beta+\delta y}\ln\left[\frac{c_1+p}{c_2+p^{\frac{1}{2}-\epsilon}(1-p)^{\frac{1}{2}+\epsilon}}\right]]$$

To show that the lower bound on the dependency on spurious correlations induced by MAT and LDAT does not increase monotonically with $p$ under some conditions, we take the derivative of the obtained lower bound $g(p)$ in Theorem 4.1 with respect to $p$:

$$\frac{\partial g(p)}{\partial p} := \frac{\partial \left( \frac{c_1+p}{c_2+p^{\frac{1}{2}-\epsilon}(1-p)^{\frac{1}{2}+\epsilon}} \right)}{\partial p}$$

Since the denominator of $\frac{\partial g(p)}{\partial p}$ is positive, we pick out the numerator: $c_2 + (\frac{1}{2}+\epsilon)p^{\frac{1}{2}-\epsilon}(1-p)^{\frac{1}{2}+\epsilon} - c_1(\frac{1}{2}-\epsilon)p^{-\frac{1}{2}-\epsilon}(1-p)^{\frac{1}{2}+\epsilon} + c_1(\frac{1}{2}+\epsilon)p^{\frac{1}{2}-\epsilon}(1-p)^{-\frac{1}{2}+\epsilon} + (\frac{1}{2}+\epsilon)p^{\frac{3}{2}-\epsilon}(1-p)^{-\frac{1}{2}+\epsilon}$. In order to study the positive and negative change of the numerator, we continue to derive it with respect to $p$ and obtain

$$p^{-\frac{3}{2}-\epsilon}(1-p)^{-\frac{3}{2}+\epsilon}[(\frac{1}{4}-\epsilon^2)(p+c_1)].$$

Since we assume $\beta > |\delta|$ and $c_1 > 0$, $\frac{\partial^2 g(p)}{\partial p^2} > 0$ and $\frac{\partial g(p)}{\partial p}$ increase with $p$ monotonically. The minimum of $\frac{\partial g(p)}{\partial p}$ is reached when $p = \frac{1}{2}$. This minimum equals to $2\epsilon c_1 + c_2 + \frac{3}{4} + \frac{3}{2}\epsilon$. When $2\epsilon c_1 + c_2 + \frac{3}{4} + \frac{3}{2}\epsilon < 0$, the lower bound does not increase with $p$ monotonically when p is within a certain range $(\in (0.5, 1))$.

$\square$

## B  Detailed Statement of Theorem 2 in Work of Nagarajan, et al.

We now introduce the Theorem 2 in [2]. Before introducing it, we first introduce the concept of the *easy-to-learn tasks* in [2], i.e. tasks with a set of constraints. The motivation of restricting ourselves to the constrained set of tasks is that it prevents us from designing complex examples where ERM is forced to rely on spurious features due to a not-so-fundamental factor. Each constraint forbids a specific failure mode of ERM in OOD scenarios. The Theorem 2 in [2] shows that even under such favorable conditions for ERM, this classical method can also be perturbed by the spurious features.

**Notations.**  For convenience, we will give some notations here again. Consider an input space $\mathcal{X}$ and a label space $\mathcal{Y} \in \{-1, 1\}$. Let $\mathcal{D} \in \mathbb{D}$ denote a distribution over $\mathcal{X} \times \mathcal{Y}$. $p_{\mathcal{D}}$ denotes the probability density function (PDF) of $\mathcal{D}$. Let $\mathbb{H}$ denote a class of classifiers $h : \mathcal{X} \to \mathbb{R}$. Consider a dataset $S$ drawn from $\mathcal{D}$. Let $L_{\mathcal{D}}(h) := \mathbb{E}_{(x,y)\sim\mathcal{D}}[h(x) \cdot y < 0]$ the loss of $h$ on $\mathcal{D}$. Let $h^\star = \arg\min_{h\in\mathbb{H}}\max_{\mathcal{D}\in\mathbb{D}}L_{\mathcal{D}}(h)$ denote the optimal classifier in the worst case. With an abuse of notation, we also denote the PDF of the distribution over $\mathcal{X}_{\text{inv}} \times \mathcal{X}_{\text{sp}}$ as $p_{\mathcal{D}}(\cdot)$. Let $\mathcal{D}_{\text{train}}$ denote the distribution of the pooled training data. Assume that there exists a mapping $\Phi : \mathcal{X}_{\text{inv}} \times \mathcal{X}_{\text{sp}} \to \mathcal{X}$ such that each $\mathcal{D} \in \mathbb{D}$ is induced by a distribution over $\mathcal{X}_{\text{inv}} \times \mathcal{X}_{\text{sp}}$.

**Definition B.1.  Easy-to-learn tasks**. Tasks that satisfy the following constraints are easy-to-learn.

1. **Fully predictive invariant features.** For all $\mathcal{D} \in \mathbb{D}$, $L_{\mathcal{D}}(h^\star) = 0$.

2. **Identical invariant distribution.** Across all $\mathcal{D} \in \mathbb{D}$, $p_{\mathcal{D}}(x_{\text{inv}})$ is identical.

3. **Conditional independence.** For all $\mathcal{D} \in \mathbb{D}$, $x_{\text{sp}} \perp x_{\text{inv}}$.

4. **Two-valued spurious features.**  We set $x_{\text{sp}} = \mathbb{R}$ and the support of $x_{\text{sp}}$ in $D_{\text{train}}$ is $\{-\beta, +\beta\}$.

5. **Identity mapping.** $\Phi$ is the identity mapping i.e., $x = (x_{\text{inv}}, x_{\text{sp}})$.

**Theorem B.2.**  *(The Theorem 2 in [2]) Let H be the set of linear classifiers $h(x) = w_{inv} \cdot x_{inv} + w_{sp}x_{sp}$. Then, for any easy-to-learn task, continuous-time gradient descent training of $w_{inv}(t)x_{inv} + w_{sp}(t)x_{sp}$ to minimize the exponential loss, satisfies:*

$$\Omega(\frac{\ln\frac{c+p}{c+\sqrt{p(1-p)}}}{M\ln(t+1)}) \leq \frac{w_{sp}(t)\beta}{|w_{inv} \cdot x_{inv}|} \leq \mathcal{O}(\frac{\ln\frac{p}{1-p}}{\ln(t+1)}) \tag{13}$$

*where $M = \max_{x\in S}\hat{w}x$ where $\hat{w}$ is the max-margin classifier on S. $c := \frac{2(2M-1)}{\beta^2}$.*

Table 4: Hyperparameter setting of the experiment of Table 1, Figure 1, Figure 5, Table 7 and Table 8.

| Parameter | Value |
|---|---|
| learning rate $r$ | 0.00005 |
| batch size $b$ | 64 |
| weight decay | 0.001 |
| drop out | 0.1 |
| AT perturbation radius $\epsilon$ (excluding Figure 1) | 0.1 |
| FGSM step size $\gamma$ (excluding Figure 1) | 0.1 |
| perturbation weight $\alpha$ learning rate $\eta$ (MAT) | 0.001 |
| factor matrix $A$ ($B$) learning rate $\rho$ (LDAT) | 0.01 |

## C Experiment Details and Supplementary Experimental Results

### C.1 Settings of the Toy Experiments

For the experiment in Table 1, Figure 1, Figure 5, Table 7, and Table 8, we use a fixed set of hyperparameters (see Table 4) instead of conducting a random search of 20 trials over the hyperparameter distribution (the setting in [9]) for efficiency. We report the average across three independent runs. For model selection method, training-domain validation [9] is used for PACS, OfficeHome, and VLCS. For NICO, an OOD validation set is adopted following [21].

### C.2 Experiment Setting and Additional Results of Table 2

**Overall setup.** We conduct a random search of 8 trials for PACS, OfficeHome, VLCS and 6 random trials for NICO and Colored MNIST in the hyperparameter search space, instead of 20 trials adopted in [9] for feasibility. We then average the best results for each hyperparameter combination and dataset (according to each model selection criterion) across test domains (except for Colored MNIST where we test on one biased domain only). Finally, we report the average of this number across three independent runs, and its corresponding standard error. We run all datasets for 8000 epochs during the training process.

**Hyperparameter Search Space.** We use a smaller hyperparameter search space than that in [9]. The search space for PACS, OfficeHome, VLCS, NICO and Colored MNIST is shown in Table 5. To determine the search space of a hyperparameter for benchmark running, we first fix other parameters and conduct a grid search to determine the approximate range of the better performances. Take the learning rate $\eta$ of the MAT matrices as an example, we fix $k = 20$ and try different values of $\eta$ on PACS. We find that the results of $\eta = 0.01$ ($82.2 \pm 0.4\%$) and $\eta = 0.001$ ($82.3 \pm 0.5\%$) are better than that of $\eta = 0.1$ ($81.6 \pm 0.2\%$), so we adopt the random search space of $\{0.01, 0.001\}$. The same is true for the other parameters.

In practical applications, as for the choice of the optimal parameter, we find that through experiments that for MAT and LDAT, the value of the rank $k$ (MAT) and $l$ (LDAT) with good test accuracy (outperforms ERM) on one data set also has good one on other datasets, as shown in the Table 6. In Table 6, $k = 10$ for MAT and $l = 15$ for LDAT outperform ERM on all three datasets, as marked in **bold**. Thus, we could find an optimal set of parameters with the model selection methods and then apply them to other datasets.

**Model Selection Stategy.** For PACS, OfficeHome and VLCS datasets, we use training-domain validation proposed in [9]. This model selection method first randomly collect 20% of each training domain to form a validation set. Then, it chooses the hyperparameter maximizing the accuracy on the validation set. For NICO, we adopt the OOD validation proposed in [21]. This method chooses the model maximizing the accuracy on a validation set that follows neither the distribution of the training domain or the distribution of the test domain. For Colored MNIST, we use test-domain validation, i.e., using a validation set that follows the distribution of the test domain. This is because it can enlarge the gaps in OOD performance among the algorithms while the gap induced by training-domain validation on Colored MNIST is marginal.

Table 5: Hyperparameter setting of the experiment on PACS, OfficeHome, VLCS, NICO and Colored MNIST of Table 2

| Dataset | Parameter | Value |
|---|---|---|
| PACS, OfficeHome, VLCS | learning rate $r$ | 0.00005 |
| | batch size $b$ | 64 |
| | weight decay (ERM, AT) | $10^{\text{Uniform}(-4,-3)}$ |
| | weight decay (MAT, LDAT) | 0.001 |
| | drop out (ERM, AT) | RandomChoice([0,0.1,0.5]) |
| | drop out (MAT, LDAT) | 0.1 |
| | perturbation number $k$ (MAT) | RandomChoice([5,10,15,20]) |
| | perturbation weight $\alpha$ learning rate $\eta$ (MAT) | RandomChoice([0.01,0.001]) |
| | perturbation rank $l$ (LDAT) | RandomChoice([5,10,15,20]) |
| | factor matrix $A$ ($B$) learning rate $\rho$ (LDAT) | RandomChoice([0.1,0.01]) |
| NICO | learning rate $r$ | 0.00005 |
| | batch size $b$ | 64 |
| | weight decay | $10^{\text{Uniform}(-4,-3)}$ |
| | drop out | RandomChoice([0,0.1,0.5]) |
| | perturbation number $k$ (MAT) | $\text{Uniform}(10,20)$ |
| | perturbation weight $\alpha$ learning rate $\eta$ (MAT) | 0.001 |
| | perturbation rank $l$ (LDAT) | $\text{Uniform}(10,20)$ |
| | factor matrix $A$ ($B$) learning rate $\rho$ (LDAT) | 0.01 |
| Colored MNIST | learning rate $r$ | $10^{\text{Uniform}(-4.5,-3.5)}$ |
| | batch size $b$ | $2^{\text{Uniform}(3,9)}$ |
| | weight decay | 0 |
| | drop out | RandomChoice([0,0.1,0.5]) |
| | perturbation number $k$ (MAT) | $\text{Uniform}(5,20)$ |
| | perturbation weight $\alpha$ learning rate $\eta$ (MAT) | $10^{\text{Uniform}(-3,-2)}$ |
| | perturbation rank $l$ (LDAT) | $\text{Uniform}(10,20)$ |
| | factor matrix $A$ ($B$) learning rate $\rho$ (LDAT) | 0.01 |
| | AT perturbation radius $\epsilon$ (MAT, LDAT) | $10^{\text{Uniform}(-1,2)}$ |
| | FGSM step size $\gamma$ (MAT) | $10^{\text{Uniform}(-2,1)}$ |
| | FGSM step size $\gamma$ (AT) | 0.1 |
| All except Colored MNIST | AT perturbation radius $\epsilon$ | 0.1 |
| | FGSM step size $\gamma$ (AT, MAT) | 0.1 |

Table 6: The test accuracy (%) on PACS, VLCS and NICO with different $k$ and $l$. The ERM baseline on these datasets is $79.7 \pm 0.4$, $74.2 \pm 1.0$, $69.7 \pm 1.0$ respectively.

| Dataset | Algorithm | $k$ or $l = 5$ | $k$ or $l = 10$ | $k$ or $l = 15$ | $k$ or $l = 20$ | $k$ or $l = 25$ | $k$ or $l = 30$ |
|---|---|---|---|---|---|---|---|
| PACS | MAT | $80.6 \pm 0.8$ | $\mathbf{80.9 \pm 0.2}$ | $81.6 \pm 0.3$ | $82.3 \pm 0.5$ | $80.8 \pm 0.1$ | $80.9 \pm 0.4$ |
| | LDAT | $79.4 \pm 0.5$ | $82.2 \pm 0.6$ | $\mathbf{80.1 \pm 0.4}$ | $79.9 \pm 0.4$ | $80.0 \pm 0.5$ | $81.2 \pm 0.4$ |
| VLCS | MAT | $74.2 \pm 0.8$ | $\mathbf{74.6 \pm 0.5}$ | $74.4 \pm 0.6$ | $74.4 \pm 0.2$ | $72.9 \pm 0.3$ | $74.4 \pm 0.6$ |
| | LDAT | $74.0 \pm 0.3$ | $74.4 \pm 0.1$ | $\mathbf{75.3 \pm 0.5}$ | $75.0 \pm 0.5$ | $74.1 \pm 0.4$ | $74.2 \pm 0.7$ |
| NICO | MAT | $69.8 \pm 1.3$ | $\mathbf{70.5 \pm 1.2}$ | $71.1 \pm 1.3$ | $69.5 \pm 2.7$ | $71.8 \pm 1.5$ | $69.3 \pm 0.7$ |
| | LDAT | $66.2 \pm 1.7$ | $67.7 \pm 0.3$ | $\mathbf{70.0 \pm 1.1}$ | $67.8 \pm 2.0$ | $68.0 \pm 1.3$ | $67.2 \pm 1.5$ |

**Backbone Network.** We use ResNet-18 [20] pretrained on ImageNet [35] for PACS, OfficeHome and VLCS. We use unpretrained ResNet-18 for NICO since it contains images largely overlapped with ImageNet classes. As for Colored MNIST, We use a small CNN-architecture following [9].

**Impact of Learning Rate.** We further investigate the impact on OOD performances of the learning rate for the perturbation weights in MAT and the learning rate for the decomposed factors in LDAT. We use the experimental setting introduced in Appendix C.1. The results of MAT and LDAT are shown in Table 7 and 8 respectively. In Table 7 and 8 we observe that the learning rate for the

Table 7: The test accuracy (%) on PACS of MAT when the learning rate ($\eta$) for the perturbation weights takes different values. We set the number of perturbations $k = 20$. The other hyperparameters take value in Table 4.

| | | MAT | | |
|---|---|---|---|---|
| ERM | AT | $\eta = 0.1$ | $\eta = 0.01$ | $\eta = 0.001$ |
| $79.7 \pm 0.0$ | $81.5 \pm 0.4$ | $81.6 \pm 0.2$ | $82.2 \pm 0.4$ | $82.3 \pm 0.5$ |

Table 8: The test accuracy (%) on PACS of LDAT when the learning rate ($\rho$) for the decomposed factors takes different values. We set the rank of perturbations $l = 10$. The other hyperparameters take value in Table 4.

| | | LDAT | |
|---|---|---|---|
| ERM | AT | $\rho=0.1$ | $\rho = 0.01$ |
| $79.7 \pm 0.0$ | $81.5 \pm 0.4$ | $82.2 \pm 0.6$ | $82.6 \pm 0.2$ |

perturbations has a marginal effect on the OOD accuracy. Both MAT and LDAT outperform ERM and AT on PACS when using different values of learning rate.

### C.3 Comparing to Existing Data Augmentation Baseline

To better verify the improvement of our proposed method on the existing adversarial augmentation methods for OOD, we reproduce the algorithm in [5]. [5] proposed a minimax iterative training procedure to generate adversarial data that follows fictitious target distributions (GUT). As discussed in Section 2, their work is restricted in the framework of using Wasserstein distance to measure the distribution shift, which is less practical for the real-world OOD setting where domain shifts are diverse. Additionally, They focus only on sample-wise operations and ignore the use of common features within a domain. The experimental results on NICO dataset is in Table 9. The unique hyperparameters of GUT follow the Settings in [5] except we set the $T_{max}$ to be 5 instead of 15 for efficiency. We can see that both our proposed method outperform GUT.

### C.4 Comparing to Existing Data Augmentation Baseline under Single-training Domain Generalization Setting

We conduct experiments to further verify the effectiveness of MAT and LDAT under single-training domain generalization setting, i.e., using only one training domain and generalize on the others. We compare our work with Neuron Coverage-Guided Domain Generalization (NCDG) [36]. The results are in Table 10. Both MAT and LDAT outperform NCDG under the scenario of single-source domain generalization.

## D Detailed Description of LDAT

In this section, we describe the detailed training procedure of LDAT (see Algorithm 2). We conduct a single-step gradient ascent for the inner maximization for the perturbations LDAT. We adopt $l_2$ norm for the perturbations.

Table 9: The test accuracy (%) on NICO.

| ERM | MAT | LDAT | GUT |
|---|---|---|---|
| $73.2 \pm 1.9$ | $\mathbf{74.2 \pm 1.5}$ | $74.4 \pm 1.6$ | $66.6 \pm 1.7$ |

Table 10: The test accuracy (%) of single-training domain generalization on PACS. Each line represents a case when we train on one domain and test on the other domains. '-' means we train on this domain, and test on the other three domains.

| Algorithm | A | C | P | S | avg |
|---|---|---|---|---|---|
| MAT | - | 73.8 | 94.1 | 74 | 80.6 |
|  | 78.5 | - | 94.2 | 75.9 | 82.9 |
|  | 80.9 | 73.7 | - | 76.6 | 77.1 |
|  | 80.4 | 76.3 | 93.3 | - | 83.3 |
| LDAT | - | 74.8 | 94.2 | 75.8 | 81.6 |
|  | 77.2 | - | 93.9 | 75.6 | 82.3 |
|  | 78.5 | 77.9 | - | 80.4 | 79 |
|  | 74.3 | 76.4 | 94.7 | - | 81.8 |
| NCDG | - | 68.6 | 95.0 | 66.4 | 76.6 |
|  | 71.6 | - | 85.8 | 71.9 | 76.4 |
|  | 68.8 | 29.8 | - | 48.6 | 49.0 |
|  | 45.6 | 65.8 | 47.9 | - | 53.1 |

---

**Algorithm 2** Detailed Training Procedure of LDAT

---

**Require:**
   Labeled training data of $m$ domains $E_1, ..., E_m$, where $E_e := \{(x_i^e, y_i^e)\}_{i=1}^{n_e}$,
   rank of the perturbations $l$, factor $A$, $B$ learning rate $\rho$,
   FGSM step size $\gamma$, perturbation radius $\epsilon$,
   number of training epochs $T$, learning rate for model parameters $r$, batch size $b$.
**Ensure:**
   Updated model $f_\theta$ with parameter $\theta$.
1:  Randomly initiate $\theta$, perturbation $\delta^e$, factor $A^e$, $B^e$ such that $||\delta^e||_2 \leq \epsilon, \forall e \in \{1, ..., m\}$.
2:  **for** iterations in $1, 2, ..., T$ **do**
3:    **for** $e$ in $1, 2, ..., m$ **do**
4:       Randomly select batch $\mathcal{B}^e = \{(x_u^e, y_u^e)\}_{u=1}^b$ from domain $E_e$.
5:       Compute the adversarial sample: $x_u^{e'} = x_u^e + A^e B^e, \forall u \in \{1, ..., b\}$
6:       Update $A^e$ by $A^e \leftarrow A^e + \rho \frac{1}{b} \sum_{u=1}^b \nabla_{A^e} \mathcal{L}(f_\theta(x_u^{e'}), y_u^e), \forall u \in \{1, ..., b\}$.
7:       Update $B^e$ by $B^e \leftarrow B^e + \rho \frac{1}{b} \sum_{u=1}^b \nabla_{B^e} \mathcal{L}(f_\theta(x_u^{e'}), y_u^e), \forall u \in \{1, ..., b\}$.
8:       Project $\delta_i^e$ to the $l_2$ ball of radius $\epsilon$.
9:       Compute the adversarial sample: $x_u^{e'} = x_u^e + A^e B^e, \forall u \in \{1, ..., b\}$
10:      Update model parameter: $\theta \leftarrow \theta - r \frac{1}{b} \sum_{u=1}^b \nabla_\theta \mathcal{L}(f_\theta(x_u^{e'}), y_u^e), \forall u \in \{1, ..., b\}$.
11:    **end for**
12: **end for**

---