# OpenReview forum: "Improving Out-of-Distribution Generalization by Adversarial Training with Structured Priors"
_NeurIPS.cc/2022/Conference — NeurIPS 2022 Accept_

### Official Review · Reviewer_mRzf · 2022-07-10

**Rating:** 6
**Confidence:** 2
**Soundness:** 3 good
**Presentation:** 3 good
**Contribution:** 3 good

**Summary:**

This paper proposes to utilize adversarial training to improve the test accuracy on the OOD dataset. They present the low-rank perturbations to perform adversarial training, inspired by the  better performance of universal adversarial training. Experiments on benchmark datasets veriy the effectiveness.

**Questions:**

The aussmption of low-rank structure should be detailed discussed in the rebuttal.

**Ethics Review Area:**

["Privacy and Security (e.g., consent)"]

**Limitations:**

The aussmption of low-rank structure should be detailed discussed in the rebuttal.

**Strengths And Weaknesses:**

Strengths:
1. Adversarial attack for good is an interesting topic. In this paper, it utilizes the adversarial training to improve test accuracy of OOD dataset. I think this topic should be encouraged.
2. The paper gives an empirical and theoretical demonstration for the proposed method. I think it is good
3. Experiments compared with many SOTA methods verify the better performance.

Weakness:
1. My main concern lies in the assumption that because the OOD dataset (or universal perturbation) usually shows low-rank structure, inspired by this, the authors propose the low-rank adversarial perturbations to perform attacks. Actually, in the real images, many scenes are not low-rank. Therefore, I don't know whether the proposed method still works in this situation. Authors should give some examples and discuss it.

---

> ### Author Response · Authors · 2022-08-02
> **Response to Reviewer mRzf**
>
> We thank reviewer mRzf for appreciating the novelty and solidness of our work. Now we address your main concerns in the following points.
>
>  **Q1:** My main concern lies in the assumption that because the OOD dataset (or universal perturbation) usually shows low-rank structure, inspired by this, the authors propose the low-rank adversarial perturbations to perform attacks. Actually, in the real images, many scenes are not low-rank. Therefore, I don't know whether the proposed method still works in this situation. Authors should give some examples and discuss it.
>
> **A1:** We will address your concern from two aspects.
>
> 1. In OOD datasets, the data in a domain usually has similar domain-related spurious features (for example, the PACS/VLCS/NICO datasets). The same observation has been made in [1]. Even the spurious features within one image do not show clear low-rank structure, our methods can still capture the common domain-related spurious features. This is because MAT and LDAT use a same perturbation for all samples in a domain. Since the spurious features (style/background, etc.) are common in a domain and the causal features (shape, etc.) are diverse, a common perturbation will help to perturb the spurious features.  [1] [2] applied similar ideas in their works to eliminate the style information and improve generalization ability. They focus on perturbing the style information of training domains to reduce the bias towards it of the CNN models. Different from our work, they mainly exploit the diversity of the spurious features from different training domains to eliminate the spurious features. However, we achieve this by using the consistency of spurious features of different samples in the same domain.
>
> 2. The OfficeHome, "Photo" subset of PACS, and VLCS are real-world datasets that contains non-low-rank images. For example, the VOC2007 subset of VLCS has rather complicated backgrounds including indoor and outdoor scenes. Our algorithms have outperformed most baseline methods on these datasets. Although these complicated scenes do not show low-rank repetitive structures as obvious as that of pure grassland and snow field, there still exists low-rank structures that can be identified and eliminated by MAT. One example is the second row of Fig. 3. The background looks complex, but there are trees appearing in parallel. This is very common in real-world images, such as the pattern on flat wall paper, stairs and furniture with regular structures. Even in messy objects, there are recurring low-rank structures within each object.
>
> [1]Domain Generalization with MixStyle, https://arxiv.org/abs/2104.02008
>
> [2]Reducing Domain Gap via Style-Agnostic Networks, https://dgyoo.github.io/papers/arxiv19.pdf

---

> > ### Comment · Reviewer_mRzf · 2022-08-09
> > **Thanks for the response**
> >
> > Thanks for the response from authors, I don't change my score.

---

> ### Author Response · Authors · 2022-08-06
> **Need further clarification?**
>
> Thanks very much for your constructive and detailed comments. We have tried our best to address the concerns and revised our paper accordingly. Is there any unclear point that we should/could further clarify?

---

### Official Review · Reviewer_RPCG · 2022-07-11

**Rating:** 5
**Confidence:** 3
**Soundness:** 3 good
**Presentation:** 3 good
**Contribution:** 3 good

**Summary:**

The paper highlights the severe performance degradation of sample-wise AT  in OOD tasks due to the existence of much larger perturbations in OOD tasks. And the paper shows a theoretical perspective to the stable generalization performance of universal AT, discovering that low-dimensional domain-wise perturbation benefits removing the low-rank structured background information. Inspired by this, the paper proposes two specific constructions of low-rank structured perturbations along the sample number and input scale dimension respectively, and theoretically proves their enhancements to the OOD robustness using a simple OOD task. To verify the approaches, experiments on five benchmark datasets are conducted to show the OOD performance of proposed MAT and LDAT compared to existing algorithms.

**Questions:**

nil

**Limitations:**

nil

**Strengths And Weaknesses:**

Strengths:

1.  The paper provides an empirical insight into the limitations of sample-wise AT on OOD tasks, and highlights the importance of domain-wise AT. Then, two effective low-rank based domain-wise AT methods are proposed. Superior performance is observed on the OOD generalization ability comparing to baselines.

2.  The paper reveals some relationship between adversarial training and OOD generalization problem theoretically, allowing for deeper dive into the AT theory.



Weaknesses:


1. The theoretical results that reveal the low-rank perturbation can prevent the model from learning spurious features is obscure. It is better to show the results in a more friendly way.

2. Some experimental results are not convincing. For instance, in Table 2, if not considering the colored MNIST datasets, the performance of the proposed model is on par with ERM. Without considering colored MNIST, the averaged gain is very limited.

---

> ### Author Response · Authors · 2022-08-02
> **Response to Reviewer RPCG**
>
> Thank you for your positive comments on this work! We address your main concerns as follows.
>
> **Q1:** The theoretical results that reveal the low-rank perturbation can prevent the model from learning spurious features is obscure. It is better to show the results in a more friendly way.
>
> **A1:** Thank you for your valuable advice. We would like to provide some explanations on our theoretical results. The Theorem 4.1 proves the effectiveness of MAT and LDAT according to the following logic.
>
> 1. As stated in the the first paragraph of Section 4, reducing the number of the perturbations from the number of samples $n_e$ to 1 is a way to reduce the rank of the perturbation. Therefore, we consider using a same perturbation for the samples in a domain as a way of low-rank adversarial training.
> 2. MAT and LDAT use domain-wise perturbations, i.e., both methods use a same perturbation for all samples in one training domain.
> 3. Theorem 4.1 illustrates the role of domain-wise perturbations in avoiding reliance on spurious features. Specifically, when adding a perturbation to the samples in a training domain,  the lower bound of the dependence of the classifier on the spurious feature $\frac{w_{sp}(t)\beta}{w_{inv}(t)x_{inv}}$ , is $\frac{\frac{1}{\beta- \delta y}\ln[\frac{c_1+p}{c_2+p^{\frac{1}{2}+\epsilon}(1-p)^{\frac{1}{2}-\epsilon}}]}{M\ln(t + 1)}$ . From the analysis in Appendix A, we know that this lower bound does not increase monotonically with the ratio of the spurious correlation $p$. This means when $p$ gets larger, model trained by MAT or LDAT will learn less spurious features. However, if we remove the domain-wise perturbations, that is, using ERM, the lower bound will be $\frac{\ln \frac{1+p}{1+\sqrt{p(1-p)}}}{\ln t}$. Further, [2] states that this lower bound $\frac{\ln \frac{1+p}{1+\sqrt{p(1-p)}}}{\ln t}$ grows monotonically with $p$. Thus, we can draw the conclusion that domain-wise perturbations are helpful to reduce dependence on spurious features comparing to ERM. Take the simple OOD task in Section 4.3 as an example. When the ratio of the cow class to the camel class are both 1:1 in the grass domain and the desert domain (i.e. $p=0.5$), there are no spurious correlations in the training set. When $p=0.8$, the ratio of the cow class to camel class is 4:1 in the grass domain and the 1:4 in the desert domain. Our theoretical results reveal that even when $p=0.8$, MAT/LDAT will not be fooled by the background-label correlation while ERM will.
> 4. Thus, MAT and LDAT can avoid using spurious features and improve OOD performance.
>
> **Q2:** Some experimental results are not convincing. For instance, in Table 2, if not considering the colored MNIST datasets, the performance of the proposed model is on par with ERM. Without considering colored MNIST, the averaged gain is very limited.
>
> **A2:** We now give the average test accuracy on the other datasets except for Colored MNIST.
>
> The Table (a) and (b) below show the average test accuracy (%) on PACS, OfficeHome and NICO. We can see that among all the algorithms, only MAT and LDAT outperform ERM. Even excluding Colored MNIST, our methods still outperform ERM by **0.4~1.4%**, whereas other methods show **no improvement** over ERM. From these results, we can see that **the promotion of our  methods is higher than the other works, and our methods clearly outperform ERM.**
>
> Table (a): average test accuracy on PACS, OfficeHome and NICO of ERM (Our runs), AT (Our runs), MAT and LDAT.
>
> | ERM (Our runs) | AT (Our runs) | MAT      | LDAT |
> | -------------- | ------------- | -------- | ---- |
> | 72.3           | 71.5          | **73.7** | 72.7 |
>
> Table (b): average test accuracy on PACS, OfficeHome and NICO of other existing methods.
>
> | ERM  | RSC  | MMD  | SagNet | COARL | IRM  | VREx | GroupDRO | DANN | MTL  | Mixup | ANDMask | MLDG |
> | ---- | ---- | ---- | ------ | ----- | ---- | ---- | -------- | ---- | ---- | ----- | ------- | ---- |
> | 72.1 | 71.8 | 71.3 | 71.2   | 71.2  | 70.6 | 72.1 | 71.8     | 70.9 | 71.4 | 69.9  | 72.1    | 59.0 |
>
> The following table shows the average test accuracy on PACS, OfficeHome, NICO and VLCS. MAT and LDAT outperform ERM by 1% and 0.4% respectively.
>
> | ERM (Our runs) | AT (Our runs) | MAT      | LDAT |
> | -------------- | ------------- | -------- | ---- |
> | 72.9           | 72.7          | **73.9** | 73.3 |

---

> ### Author Response · Authors · 2022-08-06
> **Need further clarification?**
>
> Thanks very much for your constructive and detailed comments. We have tried our best to address the concerns and revised our paper accordingly. Is there any unclear point that we should/could further clarify?

---

> ### Author Response · Authors · 2022-08-07
> **Comments on the rebuttal?**
>
> Dear Reviewer RPCG,
>
> We understand that perhaps you are too busy to read the rebuttal. But since there are only two days left, we are sorry to remind you again.
>
> In our response, we have addressed your concerns on the way of revealing the low-rank perturbation can prevent learning spurious features. We explain the logic of how we draw the conclusion that MAT/LDAT can improve OOD robustness. As for your concerns on the over-promotion of the average results caused by CMNIST, we conduct further experiments and prove that our methods still beat ERM on datasets except for CMNIST in A2.
>
> We are wondering if our response satisfies you? We are happy to answer any further questions.
>
> Sincerely, Authors

---

> ### Author Response · Authors · 2022-08-09
> **Looking forward to your comments**
>
> Dear Reviewer RPCG,
>
> There are only several hours left for the discussion. Hope we could have the last chance to discuss with you! We really have spent lots of time and effort preparing a very very detailed response. Your comments are very important to us. Could you please have a look? Many thanks!
>
> Have a nice day!

---

### Official Review · Reviewer_BPEE · 2022-07-11

**Rating:** 6
**Confidence:** 3
**Soundness:** 3 good
**Presentation:** 3 good
**Contribution:** 3 good

**Summary:**

This paper introduces two methods for universal adversarial attack (as opposed to sample-wise) in order to improve the OOD-generalization of image classification CNN models. It also shows theoretically how spurious feature use is reduced through UAT as well as showing empirically how these UAT methods perform well on a variety of benchmarks.

**Questions:**

I am a bit confused as to how the parameters in the MAT/LDAT matrices are selected. Supplement algorithms 1 and 2 seem quite crucial to the function of the proposed method, and at least one of them should be in the main text. Could you give me a plaintext overview for how one of the algorithms works?

**Limitations:**

The authors do not mention potential negative impacts. A small paragraph could be included to describe how classifiers can be misused.

The weaknesses described above are not particularly referenced either.

**Strengths And Weaknesses:**

Strengths:

The proposed method is novel, interesting, and relatively simple.

The paper is generally well written and easy to follow. I think the theoretical analysis is of interest to the broader community on how to apply UAT style training.

The experimental setup is comprehensive.


Weaknesses:

Unfortunately, the empirical results do not show a sufficient improvement over other baseline methods. The inclusion of colored mnist seriously alters the average, which is to be expected as the proposed method directly attacks only the color channels. When k=20, that's (in some sense) 20 different colors to be applied to each image. I do not find the colored mnist dataset to be generally insightful, therefore when I look at the other datasets, there is not a consistent improvement over other baselines. Normal AT, ERM, MAT and LDAT each get fairly good results (and often not outside the error margin). It is not clear to me when I should use MAT, LDAT or AT.

Minor:
Figure 2 has some typos (LRAT), and is a bit hard to parse. Having the blue arrow represent the entire NxN image is not intuitive. Perhaps turning the flat squares into 3d shapes would help.


Post rebuttal:
Thanks authors. While I would preferred LDAT or MAT to have a clear choice when to use one or the other, the rebuttal is sufficient reasoning. Overall, I agree with reviewer VkPQ, I have upgraded my score from 3 to 6.

---

> ### Author Response · Authors · 2022-08-02
> **Response to Reviewer BPEE (2/2)**
>
> **Q3:** Minor: Figure 2 has some typos (LRAT), and is a bit hard to parse. Having the blue arrow represent the entire NxN image is not intuitive. Perhaps turning the flat squares into 3d shapes would help.
>
> **A3:** Thank you for your careful reading and suggestions. We have fixed the typo and change the figure into 3d.
>
> **Q4:** I am a bit confused as to how the parameters in the MAT/LDAT matrices are selected.
>
> **A4:** To determine the value of a hyperparameter, we first fix other parameters and conduct a grid search to determine the approximate range of the better performances. Take the learning rate $\eta$ of the MAT matrices as an example, we fix $k=20$ and try different values of $\eta$ on PACS. We find that the results are all better when $\eta=0.01$ and $0.001$, so we adopt the random search space of $\{0.01,0.001\}$. The same is true for the other parameters.
>
> | Learning rate $\eta$ | 0.1            | 0.01           | 0.001          |
> | -------------------- | -------------- | -------------- | -------------- |
> | Test acc. (%)        | 81.6 $\pm$ 0.2 | 82.2 $\pm$ 0.4 | 82.3 $\pm$ 0.5 |
>
> **Q5:** Supplement algorithms 1 and 2 seem quite crucial to the function of the proposed method, and at least one of them should be in the main text. Could you give me a plaintext overview for how one of the algorithms works?
>
> **A5:** Due to space limitation, we did not put it in the main text for now, but we will make it in the future. We would like to give you an overview of the training process of MAT.
>
> 1. For each domain $e$, randomly initiate $k$ perturbations $\delta_j^e$ and weights $\alpha_j^e$.
> 2. For each domain $e$ of each minibatch, we calculate the linear combination of $k$ perturbations $\delta^e=\sum^k_{j=1}\alpha_j^e\delta_j^e$. Then we add this perturbation to the origin input data $x$ to obtain adversarial sample $x'$.
> 3. Calculate the classification error $\mathcal{L}(f(x'),y)$ with adversarial sample and conduct one step of gradient ascent to update $\alpha^e_j$ and $\delta_j^e$. Denote the updated ones as $\alpha^{e '}_j $ and $\delta_j^{e '}$
> 4. Obtain new adversarial sample $x''$ with the updated $\alpha$ and $\delta$, i.e. $x''=\sum^k_{j=1}\alpha_j^{e'}\delta_j^{e'}$ . Use $x''$ to conduct gradient descent to train model parameter $\theta$.
> 5. Repeat step 1~4.

---

> ### Author Response · Authors · 2022-08-02
> **Response to Reviewer BPEE (1/2)**
>
> Thanks for your valuable feedback. We address them in details as follows.
>
> **Q1:** Unfortunately, the empirical results do not show a sufficient improvement over other baseline methods. The inclusion of colored mnist seriously alters the average, which is to be expected as the proposed method directly attacks only the color channels.
>
> **A1:** We now give the average test accuracy on the other datasets except for Colored MNIST.
>
> The Table (a) and (b) below show the average test accuracy (%) on PACS, OfficeHome and NICO. We can see that among all the algorithms, only MAT and LDAT outperform ERM. Even excluding Colored MNIST, our methods still outperform ERM by **0.4~1.4%**, whereas other methods show **no improvement** over ERM. From these results, we can see that **the promotion of our proposed methods is higher than the other works, and our methods clearly outperform ERM.**
>
> Table (a): average test accuracy on PACS, OfficeHome and NICO of ERM (Our runs), AT (Our runs), MAT and LDAT.
>
> | ERM (Our runs) | AT (Our runs) | MAT      | LDAT |
> | -------------- | ------------- | -------- | ---- |
> | 72.3           | 71.5          | **73.7** | 72.7 |
>
> Table (b): average test accuracy on PACS, OfficeHome and NICO of other existing methods.
>
> | ERM  | RSC  | MMD  | SagNet | COARL | IRM  | VREx | GroupDRO | DANN | MTL  | Mixup | ANDMask | MLDG |
> | ---- | ---- | ---- | ------ | ----- | ---- | ---- | -------- | ---- | ---- | ----- | ------- | ---- |
> | 72.1 | 71.8 | 71.3 | 71.2   | 71.2  | 70.6 | 72.1 | 71.8     | 70.9 | 71.4 | 69.9  | 72.1    | 59.0 |
>
> The following table shows the average test accuracy on PACS, OfficeHome, NICO and VLCS. MAT and LDAT outperform ERM by 1% and 0.4% respectively.
>
> | ERM (Our runs) | AT (Our runs) | MAT      | LDAT |
> | -------------- | ------------- | -------- | ---- |
> | 72.9           | 72.7          | **73.9** | 73.3 |
>
>
>
> **Q2:** Normal AT, ERM, MAT and LDAT each get fairly good results (and often not outside the error margin). It is not clear to me when I should use MAT, LDAT or AT.
>
> **A2:** Since the performance of AT is not better than MAT and LDAT in our series of experiments, MAT and LDAT should be preferred. For the selection of MAT and LDAT, we provide the following two guidelines.
>
> 1. MAT is a complex and high-rank version of LDAT, which has a stronger ability to describe more complex spurious background information. The comparison of MAT and LDAT in the second row of Fig 3. indicates that when faced with a more complex background, MAT can better capture the object than LDAT does.
> 2. The training process of LDAT costs less memory than MAT, although there is no significant difference in training time between the two methods. If the memory is limited, LDAT is preferred.

---

> ### Author Response · Authors · 2022-08-06
> **Need further clarification?**
>
> Thanks very much for your constructive and detailed comments. We have tried our best to address the concerns and revised our paper accordingly. Is there any unclear point that we should/could further clarify?

---

> ### Author Response · Authors · 2022-08-07
> **Comments on the rebuttal?**
>
> Dear Reviewer BPEE,
>
> We understand that perhaps you are too busy to read the rebuttal. But since there are only two days left, we are sorry to remind you again.
>
> In our response, we have addressed your concerns on the over-promotion of the average results caused by CMNIST in A1. We conduct further experiments and prove that our methods still beat ERM on datasets except for CMNIST. For your concerns on the choice of the algorithms, we have given our suggestions in A2. We also explain the way we choose the hyperparameters in A4. As for your concerns on the description of the algorithm, we have given the detailed training procedure of MAT in A5.
> Additionally, we have taken your advice and changed the figure into 3-D.
>
> We are wondering if our response satisfies you? We are happy to answer any further questions.
>
> Sincerely, Authors

---

> ### Author Response · Authors · 2022-08-09
> **Your comments are important for us**
>
> Dear Reviewer BPEE,
>
> There are only several hours left for the discussion. We have spent lots of time and effort preparing a very very detailed response. Hope it can address your concerns. Your comments are very important to us. We are greatly appreciated if you could have a look!
>
> Have a nice day!

---

> > ### Comment · Reviewer_BPEE · 2022-08-09
> > **I'll agree with reviewer VkPQ**
> >
> > Thanks authors. While I would preferred LDAT or MAT to have a clear choice when to use one or the other, the rebuttal is sufficient reasoning. Overall, I agree with reviewer VkPQ, I have upgraded my score from 3 to 6.

---

### Official Review · Reviewer_VkPQ · 2022-07-12

**Rating:** 7
**Confidence:** 5
**Soundness:** 3 good
**Presentation:** 3 good
**Contribution:** 3 good

**Summary:**

In this submission, the authors proposed to focus on the problem of domain generalization based on adversarial training. Unlike existing adversarial training methods which inject sample-wise adversarial noise, a novel universal adversarial perturbation generating mechanism (with two variant) was proposed in this paper. Experimental results show the effectiveness of the proposed MAT and LDAT on domainbed benchmark.

**Questions:**

Please see above.


**Limitations:**

While the limitation of existing adversarial training has been discussed, the limitation of the proposed method is missed.

**Strengths And Weaknesses:**

post rebuttal

My concerns have been addressed.


Strength:

-	The proposed method is quite novel and interesting.

-	The motivation and analysis of the proposed method is strong.

-	Experimental results on benchmark justify the effectiveness of the proposed method.

Weakness:

-	My main concern is on the evaluation part. It seems that except Colored MNIST (which is quite toy), the improvement on other datasets are very limited. Moreover, the only baseline using data augmentation for comparison are conventional AT and Mixup, other adversarial augmentation for DG (e.g., GUT [Volpi, NIPS’18]) and conventional data augmentation methods (which have been shown to be quite effective [French, ICLR’18], [Tian, TPAMI’22]) should be discussed.

-	 In Fig. 4, it seems that the number of pertubations adopted between MAT and LDAT is not consistent. I am wondering if there are some guidance to choose the optimal number given source domains and model?

---

> ### Author Response · Authors · 2022-08-02
> **Response to Reviewer VkPQ (2/2)**
>
> **Q3:** In Fig. 4, it seems that the number of perturbations adopted between MAT and LDAT is not consistent. I am wondering if there are some guidance to choose the optimal number given source domains and model?
>
> **A3:** We found through experiments that for MAT and LDAT, the value of the rank $k$ (MAT) and $l$ (LDAT) with good test accuracy (outperforms ERM) on one data set also has good one on other datasets, as shown in the following tables. In these tables, $k=10$ **for MAT and $l=15$ for LDAT outperform ERM on all three datasets**, as marked in **bold**. Thus, we could find an optimal set of parameters with the model selection methods and then apply them to other datasets. The reason why the results presented here are worse than those in Table 2 in the original paper is that we use a fixed set of hyperparameters instead of searching in a small range for efficiency.
>
> Table (a): Test acc. (%) on PACS, ERM baseline is 79.7 $\pm$ 0.4
>
> | $k$ (MAT) / $l$(LDAT) | 5              | 10                 | 15                 | 20             | 25             | 30             |
> | --------------------- | -------------- | ------------------ | ------------------ | -------------- | -------------- | -------------- |
> | MAT                   | 80.6 $\pm$ 0.8 | **80.9 $\pm$ 0.2** | 81.6 $\pm$ 0.3     | 82.3 $\pm$ 0.5 | 80.8 $\pm$ 0.1 | 80.9 $\pm$ 0.4 |
> | LDAT                  | 79.4 $\pm$ 0.5 | 82.2 $\pm$ 0.6     | **80.1 $\pm$ 0.4** | 79.9 $\pm$ 0.4 | 80.0 $\pm$ 0.5 | 81.2 $\pm$ 0.4 |
>
> Table (b): Test acc. (%) on VLCS, ERM baseline is 74.2 $\pm$ 1.0
>
> | $k$ (MAT) / $l$(LDAT) | 5              | 10                 | 15                 | 20             | 25             | 30             |
> | --------------------- | -------------- | ------------------ | ------------------ | -------------- | -------------- | -------------- |
> | MAT                   | 74.2 $\pm$ 0.8 | **74.6 $\pm$ 0.5** | 74.4 $\pm$ 0.6     | 74.4 $\pm$ 0.2 | 72.9 $\pm$ 0.3 | 74.4 $\pm$ 0.6 |
> | LDAT                  | 74.0 $\pm$ 0.3 | 74.4 $\pm$ 0.1     | **75.3 $\pm$ 0.5** | 75.0 $\pm$ 0.5 | 74.1 $\pm$ 0.4 | 74.2 $\pm$ 0.7 |
>
> Table (c): Test acc. (%) on NICO, ERM baseline is 69.7 $\pm$ 1.0
>
> | $k$ (MAT) / $l$(LDAT) | 5              | 10                 | 15                 | 20             | 25             | 30             |
> | --------------------- | -------------- | ------------------ | ------------------ | -------------- | -------------- | -------------- |
> | MAT                   | 69.8 $\pm$ 1.3 | **70.5 $\pm$ 1.2** | 71.1 $\pm$ 1.3     | 69.5 $\pm$ 2.7 | 71.8 $\pm$ 1.5 | 69.3 $\pm$ 0.7 |
> | LDAT                  | 66.2 $\pm$ 1.7 | 67.7 $\pm$ 0.3     | **70.0 $\pm$ 1.1** | 67.8 $\pm$ 2.0 | 68.0 $\pm$ 1.3 | 67.2 $\pm$ 1.5 |

---

> > ### Comment · Reviewer_VkPQ · 2022-08-03
> > **Thank you**
> >
> > Thanks for the response. My concerns about experimental evaluations have been addressed. While the improvement is to some extent marginal, I think the proposed method brings merits in the community of DG and ood robustness. Therefore, I would like to raise my score to 7 (accept).
> >
> > The paper of Tian,PAMI'22 is "Neuron Coverage-Guided Domain Generalization". It would be interesting to see more different types of augmentation in the revised manuscript.

---

> > > ### Author Response · Authors · 2022-08-07
> > > **Thanks and further experiment results**
> > >
> > > Thank you for raising the score and providing the title of the paper! We further compare the performance of our methods and the approach proposed in Neuron Coverage-Guided Domain Generalization (NCDG). Since NCDG mainly aims to address the scenario of single-domain generalization, i.e., training on only one domain and test on other domains, we conduct experiments on this setting to compare the performance. The results of MAT/LDAT/NCDG are in Table (e)/(f)/(g) respectively. Each line represents a case when we train on one domain and test on the other domains. ‘-’ means we train on this domain, and test on the other three domains. We can see that MAT/LDAT clearly outperform NCDG in the single-domain generalization setting.
> > >
> > > Table (e): Test acc. (%) of single-domain generalization on PACS of MAT.
> > >
> > > | A    | C    | P    | S    | avg  |
> > > | ---- | ---- | ---- | ---- | ---- |
> > > | -    | 73.8 | 94.1 | 74   | 80.6 |
> > > | 78.5 | -    | 94.2 | 75.9 | 82.9 |
> > > | 80.9 | 73.7 | -    | 76.6 | 77.1 |
> > > | 80.4 | 76.3 | 93.3 | -    | 83.3 |
> > >
> > > Table (f): Test acc. (%) of single-domain generalization on PACS of LDAT.
> > >
> > > | A    | C    | P    | S    | avg  |
> > > | ---- | ---- | ---- | ---- | ---- |
> > > | -    | 74.8 | 94.2 | 75.8 | 81.6 |
> > > | 77.2 | -    | 93.9 | 75.6 | 82.3 |
> > > | 78.5 | 77.9 | -    | 80.4 | 79   |
> > > | 74.3 | 76.4 | 94.7 | -    | 81.8 |
> > >
> > > Table (g): Test acc. (%) of single-domain generalization on PACS of NCDG.
> > >
> > > | A    | C    | P    | S    | avg  |
> > > | ---- | ---- | ---- | ---- | ---- |
> > > | -    | 68.6 | 95.0 | 66.4 | 76.6 |
> > > | 71.6 | -    | 85.8 | 71.9 | 76.4 |
> > > | 68.8 | 29.8 | -    | 48.6 | 49.0 |
> > > | 45.6 | 65.8 | 47.9 | -    | 53.1 |

---

> > > > ### Comment · Reviewer_VkPQ · 2022-08-10
> > > > **Thanks for more results**
> > > >
> > > > It's interesting to see that the proposed method can outperform NCDG, which performs conventional data augmentation for domain generalization. I am more positive about this submission.
> > > >
> > > > Just curious (nothing to do with the decision of NeurIPS), how does the proposed method perform under the setting of data corruption? In many situations, even if the sample is not coming from a new environment, it is likely that the sample is corrupted by noise (slightly blurring for example). It would be more interesting to see if the proposed method can benefit universal data augmentation to some extent (no need to be ood), perhaps in future work.

---

> ### Author Response · Authors · 2022-08-02
> **Response to Reviewer VkPQ (1/2)**
>
> Thank you for your recognition of this paper! Here is our answer to your concern.
>
> **Q1:** My main concern is on the evaluation part. It seems that except Colored MNIST (which is quite toy), the improvement on other datasets are very limited.
>
> **A1:** We now give the average test accuracy on the other datasets except for Colored MNIST.
>
> The Table (a) and (b) below show the average test accuracy (%) on PACS, OfficeHome and NICO. We can see that among all the algorithms, only MAT and LDAT outperform ERM. Even excluding Colored MNIST, our methods still outperform ERM by **0.4~1.4%**, whereas other methods show **no improvement** over ERM. From these results, we can see that **the promotion of our proposed methods is higher than the other works, and our methods clearly outperform ERM.**
>
> Table (a): average test accuracy on PACS, OfficeHome and NICO of ERM (Our runs), AT (Our runs), MAT and LDAT.
>
> | ERM (Our runs) | AT (Our runs) | MAT      | LDAT |
> | -------------- | ------------- | -------- | ---- |
> | 72.3           | 71.5          | **73.7** | 72.7 |
>
> Table (b): average test accuracy on PACS, OfficeHome and NICO of other existing methods.
>
> | ERM  | RSC  | MMD  | SagNet | COARL | IRM  | VREx | GroupDRO | DANN | MTL  | Mixup | ANDMask | MLDG |
> | ---- | ---- | ---- | ------ | ----- | ---- | ---- | -------- | ---- | ---- | ----- | ------- | ---- |
> | 72.1 | 71.8 | 71.3 | 71.2   | 71.2  | 70.6 | 72.1 | 71.8     | 70.9 | 71.4 | 69.9  | 72.1    | 59.0 |
>
> The following table shows the average test accuracy on PACS, OfficeHome, NICO and VLCS. MAT and LDAT outperform ERM by 1% and 0.4% respectively.
>
> | ERM (Our runs) | AT (Our runs) | MAT      | LDAT |
> | -------------- | ------------- | -------- | ---- |
> | 72.9           | 72.7          | **73.9** | 73.3 |
>
>
>
> **Q2:** Moreover, the only baseline using data augmentation for comparison are conventional AT and Mixup, other adversarial augmentation for DG (e.g., GUT [Volpi, NIPS’18]) and conventional data augmentation methods (which have been shown to be quite effective [French, ICLR’18], [Tian, TPAMI’22]) should be discussed.
>
> **A2**: Following your suggestion, we have reproduced the algorithm of GUT on NICO using the same setting as in Table 2.  The unique hyperparameters of GUT follow the Settings in [Volpi, NIPS’18] except we set the $T_{max}$ to be 5 instead of 15 for efficiency. The results are as follows. Both our proposed method outperform GUT. We put these discussions in Appendix C.3.
>
> |      | Test acc. (%)  |
> | ---- | -------------- |
> | ERM  | 73.2 $\pm$ 1.9 |
> | GUT  | 66.6 $\pm$ 1.7 |
> | MAT  | 74.2 $\pm$ 1.5 |
> | LDAT | 74.4 $\pm$ 1.6 |
>
> Additionally, though very effective, the work of [French, ICLR’18] you mentioned is a Domain Adaptation algorithm that requires unsupervised data, which may not be directly applicable to OOD problems. Moreover, we are not sure which article you refer to [Tian, TPAMI '22], so we have not discussed it. We would appreciate it if you could provide us with the title of this article.

---

### Official Review · Reviewer_Nziv · 2022-07-15

**Rating:** 4
**Confidence:** 3
**Soundness:** 2 fair
**Presentation:** 3 good
**Contribution:** 2 fair

**Summary:**

This paper proposes two modified methods for adversarial training to improve robustness. The noise being added is constrained to low dimensional subspace, and it further changes to more delicate constraints.

**Questions:**

In line 245, if theorem 2 of [2] is so important, it should be explicitly written here. If the theoretical result is better than [2], then it should be specified what changes make the result better, i.e. subspace constraints, or the improvement in technical details to make inequalities tighter.

the paragraph from 202, if this is going to be a concrete example with "In the grass/desert domain, the cow/camel class predominates" specified, it might be better based on an image example showed in the paper, or not to be specified at all. I feel these examples come out of nowhere and leave for nowhere.

**Limitations:**

yes.

**Strengths And Weaknesses:**

Strength: the paper proposed a reasonable modification as there might be some subspace or clusters of pixels that are more irrelevant, so that they could be treated differently in adversarial training.

It also includes a theory analysis with simple example.

Figure 3 provides an intuitive explanation.

weakness: The adversarial training overall seem to be very unstable, whether its sample-wise in figure1, or on OfficeHome in table 2, and in figure 4. The rank of the perturbation doesn't show a reasonable correlation with test accuracy.

The theory analysis is based on a simple binary classification, which might be intuitive. But the theorem proved here, I cannot find the role of this subspace of rank K here, which should be important.

---

> ### Author Response · Authors · 2022-08-02
> **Response to Reviewer Nziv**
>
> Thank you for your positive comments and your valuable advice on this paper! Here is our answer to your concern.
>
> **Q1：** The adversarial training overall seem to be very unstable, whether its sample-wise in figure1, or on OfficeHome in table 2, and in figure 4. The rank of the perturbation doesn't show a reasonable correlation with test accuracy.
>
> **A1：**We extended the experiments on CMNIST. The results show that when the rank $k$ (MAT) and $l$ (LDAT) are large enough (analogous to the non-low-rank case of Sample-Wise), the test accuracy will decrease and become unstable. This indicates the accuracy will decrease with the increase of the rank in a wide range. The ERM baseline is 28.1 $\pm$ 1.5.
>
> |      | $k\in [5,20]$, $l\in[10,20]$ | $k$ or $l$=200  | $k$ or $l$=500 | $k$ or $l$=1000 |
> | ---- | ---------------------------- | --------------- | -------------- | --------------- |
> | MAT  | 65.4 $\pm$ 8.1               | 34.9 $\pm$ 20.2 | 25.6 $\pm$ 8.5 | 23.4 $\pm$ 10.8 |
> | LDAT | 52.5 $\pm$ 5.4               | 24.9 $\pm$ 8.9  | 19.0 $\pm$ 6.6 | 10.3 $\pm$ 0.1  |
>
> **Q2:** The theory analysis is based on a simple binary classification, which might be intuitive. But the theorem proved here, I cannot find the role of this subspace of rank K here, which should be important.
>
> **A2**: In Theorem 4.1, we prove that when adding the domain-wise perturbation to the training procedure, the lower bound of the reliance on the spurious feature will not grow monotonically with the ratio of the spurious correlation $p$. However, when simply conduct ERM, this lower bound will increase monotonically with $p$. Therefore, the domain wise perturbation (using a same perturbation for the samples in the same domain) help to improve the OOD robustness. Note that using domain-wise perturbation means that the we introduce the low-rank component. Thus, we can draw the conclusion that low-rank perturbation is benefit for OOD.
>
> **Q3:**  In line 245, if theorem 2 of [2] is so important, it should be explicitly written here. If the theoretical result is better than [2], then it should be specified what changes make the result better, i.e. subspace constraints, or the improvement in technical details to make inequalities tighter.
>
> **A3:** We have explicitly added the Theorem 2 of [2] in the Appendix B. The key to the improvement is the domain-wise perturbation, i.e., adding a same perturbation for the samples in a domain. Theorem 4.1 illustrates the role of domain-wise perturbations in avoiding reliance on spurious features. Specifically, when adding a perturbation to the samples in a training domain,  the lower bound of the dependence of the classifier on the spurious feature $\frac{w_{sp}(t)\beta}{w_{inv}(t)x_{inv}}$ , is $\frac{\frac{1}{\beta- \delta y}\ln[\frac{c_1+p}{c_2+p^{\frac{1}{2}+\epsilon}(1-p)^{\frac{1}{2}-\epsilon}}]}{M\ln(t + 1)}$ . From the analysis in Appendix A, we know that this lower bound does not increase monotonically with the ratio of the spurious correlation $p$. This means when $p$ gets larger, model trained by MAT or LDAT will learn less spurious features. However, if we remove the domain-wise perturbations, that is, using ERM, the lower bound will be $\frac{\ln \frac{1+p}{1+\sqrt{p(1-p)}}}{\ln t}$. Further, [2] states that this lower bound $\frac{\ln \frac{1+p}{1+\sqrt{p(1-p)}}}{\ln t}$ grows monotonically with $p$. Thus, we can draw the conclusion that domain-wise perturbations are **helpful to reduce dependence on spurious features comparing to ERM.**
>
> **Q4:** The paragraph from 202, if this is going to be a concrete example with "In the grass/desert domain, the cow/camel class predominates" specified, it might be better based on an image example showed in the paper, or not to be specified at all. I feel these examples come out of nowhere and leave for nowhere.
>
> **A4:** Thank you for your advice and we have added an illustration of this tiny OOD task in Section 4.3.

---

> ### Author Response · Authors · 2022-08-06
> **Need further clarification?**
>
> Thanks very much for your constructive and detailed comments. We have tried our best to address the concerns and revised our paper accordingly. Is there any unclear point that we should/could further clarify?

---

> > ### Comment · Reviewer_Nziv · 2022-08-09
> > **still feel the low rank assumption are not well defended.**
> >
> > Sorry, I failed to see the point here. as
> > " adding the domain-wise perturbation to the training procedure, the lower bound of the reliance on the spurious feature will not grow monotonically with the ratio of the spurious correlation"
> >
> > So this could be the case of making 10000-dim data perturbed with rank 9999, also counted as low rank? The analytical analysis of this rank parameter is still not clear to me.

---

> > > ### Author Response · Authors · 2022-08-09
> > > **Further Response to Reviewer Nziv**
> > >
> > > Thanks for your feedback. We would like to further elaborate our idea to address your concerns.
> > >
> > > **Q5:** Sorry, I failed to see the point here. as " adding the domain-wise perturbation to the training procedure, the lower bound of the reliance on the spurious feature will not grow monotonically with the ratio of the spurious correlation"
> > >
> > > **A5:** The conclusion "adding the domain-wise perturbation to the training procedure, the lower bound of the reliance on the spurious feature will not grow monotonically with the ratio of the spurious correlation" is drawn from the following derivation: If we simply use ERM, the lower bound of the reliance on the spurious feature is $\frac{\ln \frac{1+p}{1+\sqrt{p(1-p)}}}{\ln t}$, which grows monotonically with $p$; If we use AT with domain-wise perturbations (whether it's MAT or LDAT), the lower bound becomes $\frac{\frac{1}{\beta- \delta y}\ln[\frac{c_1+p}{c_2+p^{\frac{1}{2}+\epsilon}(1-p)^{\frac{1}{2}-\epsilon}}]}{M\ln(t + 1)}$, which does not grows monotonically with $p$.
> > >
> > > **Q6:** So this could be the case of making 10000-dim data perturbed with rank 9999, also counted as low rank?
> > >
> > > **A6:**  Particularly, for training data in a domain, the perturbation rank is approximate to $k\times H\times W \times C $ (MAT) or $l$ (LDAT), and the data dimension is $\text{Dim}=N\times H\times W \times C$ , where $N$ is the number of samples, $H/W/C$ are the size of an input sample. As we can see in the Table (a) below, when $k$ or $l$ is equal to 200/500/1000, although $k\times H\times W \times C$ and $l$ are less than $\text{Dim}$, the test accuracy is still low since they are not small enough. As stated in Section 4, there are two ways of reducing the rank, that is, reducing the number of the perturbations used in a domain and reducing the rank of a perturbation itself.  Making 10000-dim data perturbed with rank 9999 is the second way of low-rank perturbation. We can infer from the experimental results in Table (a) that since rank 9999 is not small enough, even if it's smaller than the dimension of the data, the test accuracy could be low.
> > >
> > > Table (a): Test acc. (%) on Colored MNIST with different rank of the perturbation.
> > >
> > > |      | $k\in [5,20], l\in[10,20]$ | $k$ or $l$=200  | $k$ or $l$=500 | $k$ or $l$=1000 |
> > > | ---- | -------------------------- | --------------- | :------------- | --------------- |
> > > | MAT  | 65.4 $\pm$ 8.1             | 34.9 $\pm$ 20.2 | 25.6 $\pm$ 8.5 | 23.4 $\pm$ 10.8 |
> > > | LDAT | 52.5 $\pm$ 5.4             | 24.9 $\pm$ 8.9  | 19.0 $\pm$ 6.6 | 10.3 $\pm$ 0.1  |

---

> > > > ### Comment · Reviewer_Nziv · 2022-08-09
> > > > **theory provided in the paper doesn't prove about this correlation ?**
> > > >
> > > > This experiment shows that the rank should be sufficiently small to increase test accuracy, but if the rank is too small, it will decrease accuracy. I can understand this effect, as it should have a threshold between adverse effects. But the theory provided in the paper doesn't prove about this correlation?

---

> > > ### Author Response · Authors · 2022-08-09
> > > **Further Response to Reviewer Nziv**
> > >
> > > Thanks for your question. We can empirically explain the reason why the rank that's too small or too big is harmful. When the rank is too small, it will become too trivial to capture any spurious feature since most features have non-trivial structures. When the rank is too big, it will fail to model low-rank spurious features that are common. Our theorem pointed out the importance of the role of low-rank perturbations firstly and we will leave the detailed research on the specific trend of the effect of the rank change in the future following your suggestions!

---

> ### Author Response · Authors · 2022-08-07
> **Comments on the rebuttal?**
>
> Dear Reviewer Nziv,
>
> We understand that perhaps you are too busy to read the rebuttal. But since there are only two days left, we are sorry to remind you again.
>
> In our response, we have addressed your concerns on the correlation between the rank and the test accuracy by further experiments. For your concerns on the role of the rank $k$, we demonstrate that using domain-wise perturbations (which equals the case when $k=1$) is beneficial for OOD generalization compared to ERM, therefore we validate the role of $k$ indirectly. We have also pointed out that the domain-wise perturbation is the key to better results. Finally, we have taken your advice and added an illustration of the tiny OOD task.
>
> We are wondering if our response satisfies you? We are happy to answer any further questions.
>
> Sincerely, Authors

---

### Meta-Review · Area_Chair_A58S · 2022-08-26

**Recommendation:** Accept
**Confidence:** Less certain

**Metareview:**

The authors have provided a strong rebuttal addressing many of the concerns brought up by reviewers. Two of the reviewers have in response increased their scores significantly. At the end of the process most of the reviewers have converged in favour of acceptance.

**Award:**

No

---

### Decision · Program_Chairs · 2022-09-14

Accept